# Oxygen chemoreceptor inhibition by dopamine D$_2$ receptors in isolated zebrafish gills

Maddison Reed[1] and Michael G. Jonz[1,2] 

[1]*Department of Biology, University of Ottawa, Ottawa, ON, Canada*
[2]*Brain and Mind Research Institute, University of Ottawa, Ottawa, ON, Canada*

Handling Editors: Harold Schultz & Andrew Holmes

The peer review history is available in the Supporting Information section of this article (https://doi.org/10.1113/JP287824#support-information-section).

**Abstract figure legend** Transgenic Tg(elavl3:GCaMP6s) zebrafish, which express an endogenous calcium reporter in gill oxygen chemoreceptors and nerve terminals, were used to explore the role of dopamine in hypoxia signalling. We discovered that dopamine via presynaptic dopamine D$_2$ receptors (D$_2$R) provides a feedback mechanism attenuating the chemoreceptor response to hypoxia.

**Abstract**   Dopamine is an essential modulator of oxygen sensing and control of ventilation and is the most well described and abundant neurotransmitter in the mammalian carotid body. Little is known of the evolutionary significance of dopamine in oxygen sensing, or whether it plays a similar role in anamniotes. In the model vertebrate, zebrafish (*Danio rerio*), presynaptic dopamine D$_2$ receptor (D$_2$R) expression was demonstrated in gill neuroepithelial cells (NECs), analogues of mammalian oxygen chemoreceptors; however, a mechanism for dopamine and D$_2$R in the gills had not been defined. The present study tested the hypothesis that presynaptic D$_2$Rs provide a

Maddison Reed is a PhD candidate in the Department of Biology at the University of Ottawa, Canada. Her research focuses on the neurochemicals that facilitate oxygen sensing and the underlying pathways that regulate oxygen homeostasis in vertebrates.

This article was first published as a preprint. Reed M, Jonz MG. 2024. Oxygen chemoreceptor inhibition by dopamine D$_2$ receptors in isolated zebrafish gills. bioRxiv. https://doi.org/10.1101/2024.10.08.617247

The Journal of Physiology

feedback mechanism attenuating the chemoreceptor response to hypoxia. Using an isolated gill preparation from Tg(*elavl3*:GCaMP6s) zebrafish, we measured hypoxia-induced changes in intracellular $Ca^{2+}$ concentration ($[Ca^{2+}]_i$) in NECs and postsynaptic neurons. Activation of $D_2R$ with dopamine or specific $D_2R$ agonist, quinpirole, decreased hypoxic responses in NECs; whereas $D_2R$ antagonist, domperidone, had the opposite effect. Addition of SQ22536, an adenylyl cyclase (AC) inhibitor, decreased the effect of hypoxia on $[Ca^{2+}]_i$, similar to dopamine. Activation of AC by forskolin partially recovered the suppressive effect of dopamine on the $Ca^{2+}$ response to hypoxia. Furthermore, we demonstrate that the response to hypoxia in postsynaptic neurons was dependent upon innervation with NECs, and was subject to modulation by activation of presynaptic $D_2R$. Our results provide the first evidence of neurotransmission of the hypoxic signal at the NEC-nerve synapse in the gill and suggest that a presynaptic, modulatory role for dopamine in oxygen sensing arose early in vertebrate evolution.

(Received 8 October 2024; accepted after revision 13 February 2025; first published online 8 March 2025)

**Corresponding author** M. G. Jonz: Department of Biology, University of Ottawa, 30 Marie Curie Pvt., Ottawa, ON K1N 6N5, Canada.    Email: mjonz@uottawa.ca

## Key points

- For the first time, we present an experimental model that permits imaging of intracellular $Ca^{2+}$ in identified oxygen chemoreceptors in zebrafish using GCaMP in a whole/intact sensing organ.
- The hypoxic response of zebrafish chemoreceptors is attenuated by dopamine through a mechanism involving $D_2$ receptors and adenylyl cyclase.
- Zebrafish oxygen chemoreceptors send a hypoxic signal to postsynaptic (sensory) neurons.
- Postsynaptic neuronal responses to hypoxia are modulated by presynaptic $D_2$ receptors, suggesting a link between chemoreceptor inhibition by dopamine and modulation of the hypoxic ventilatory response.
- Our results suggests that a modulatory role for dopamine in oxygen sensing arose early in vertebrate evolution.

## Introduction

Dopamine is an important neuromodulator involved in oxygen sensing and control of reflex hyperventilation. In the mammalian carotid body, it is the most abundant neurotransmitter and has been described in numerous species (Gauda, 2002). Hypoxic activation of carotid body oxygen chemosensory type 1 (glomus) cells involves $Ca^{2+}$-dependent neurotransmitter release to act on sensory terminals of the carotid sinus nerve (González et al., 1994; Kumar & Prabhakar, 2012; López-Barneo et al., 1988; Nurse, 2010). Dopamine, released by type 1 cells, has been shown to have autocrine–paracrine actions in the carotid body via G-protein-coupled dopamine $D_2$ receptors ($D_2R$) on both presynaptic type 1 cells and postsynaptic afferent terminals (Carroll et al., 2005; Gauda, 2002; Itturiaga et al., 2009; Mir et al., 1984; Nurse, 2010; Zhang et al., 2018). At the presynaptic type 1 cell, the inhibitory actions of dopamine decrease further neurotransmitter release, modulating carotid body hypoxia signalling (Benot & López-Barneo, 1990). Dopamine has

also been implicated in regulation of chemosensitivity during acclimatization to prolonged periods of hypoxia (Bisgard, 2000; Huey & Powell 2000; Powell, 2007) and as an important factor in development or maturation of the carotid body (Carroll et al., 2005; De Caro et al., 2013; Gauda et al., 1996, Gauda, 2002).

In fish, oxygen sensing occurs in the gills via chemoreceptive neuroepithelial cells (NECs) in a manner similar to carotid body type I cells. NECs are found in the gill filament epithelium and respond to a decrease in $P_{O_2}$ by inhibition of background $K^+$ channels, membrane depolarization and $Ca^{2+}$-dependent vesicular recycling (Jonz et al., 2004; Zachar et al., 2017a). The latter is consistent with neurotransmitter release into the synaptic cleft, which may lead to activation of sensory nerve fibres to control ventilation. NECs are polymodal, such that they are sensitive to changes in $O_2$, $CO_2$, $H^+$ and lactate (Abdallah et al., 2015; Jonz, 2018; Leonard et al., 2022; Qin et al., 2010) and are characterized by immunoreactivity to serotonin (5-HT) and synaptic vesicle protein, SV2 (Jonz & Nurse, 2003).

Teleost gill arches are innervated by branches of the glossopharyngeal (IX) and vagus (X) cranial nerves, which carry parasympathetic efferent fibres to the gill vasculature and sensory (afferent) fibres from chemoreceptors of the gill filaments (de Graaf, 1990; Nilsson, 1984; Sundin & Nilsson, 2002). Beneath the efferent filament artery lies a series of four to six neuronal somata found along a nerve bundle that travels the length of the filament. These are referred to as chain neurons (ChNs) and may provide an additional source of sensory innervation to NECs (Jonz & Nurse, 2003).

The gill arches in fish are homologues of the sites of O$_2$-sensing organs in mammals, such as the carotid body and pulmonary neuroepithelial bodies; however, whether gill NECs are functionally similar (i.e. homologous) with carotid body type I cells or neuroepithelial bodies, is controversial (Hockman et al., 2017; Jonz, 2024; Milsom & Burleson, 2007; Zachar & Jonz, 2012). Despite the similarities between NECs and type I cells, and in contrast to the well-described neurochemistry of the carotid body (Nurse, 2010), a mechanism for control of ventilation during hypoxia via neurotransmission in the gills of fish has never been defined.

Early evidence of a role for dopamine in the gills was shown in isolated gills of rainbow trout (*Oncorhynchus mykiss*), where dopamine caused a small, brief burst in afferent nerve activity followed by mild inhibition (Burleson & Milsom, 1995). In live, whole-animal experiments using zebrafish larvae, application of dopamine or the dopamine D$_2$R agonist, quinpirole, decreased ventilation frequency, suggesting inhibitory effects of dopamine on ventilation (Reed et al., 2024; Shakarchi et al., 2013). In isolated gill tissue from adult zebrafish, quantitative PCR analysis confirmed expression of *drd2a* and *drd2b* (genes encoding D$_2$Rs) in the gills, and their relative abundance decreased following acclimation to hypoxia for 48 h (Reed et al., 2024). Furthermore, immunohistochemical labelling in the gill localized D$_2$R to presynaptic NECs, as well as provided evidence for the synthesis and storage of dopamine by nerve terminals of postsynaptic sensory neurons that innervate NECs (Reed et al., 2024). Although these findings point to an inhibitory role for dopamine via D$_2$R in the gill, a direct mechanism with respect to how dopamine may participate in oxygen sensing has not been elucidated.

The present study aimed to delineate a mechanism by which presynaptic D$_2$Rs provide a feedback mechanism that attenuates the chemoreceptor response to hypoxia. Using a genetically-encoded Ca$^{2+}$ indicator in gills isolated from transgenic zebrafish, we found that activation of presynaptic D$_2$R decreased the NEC Ca$^{2+}$ response to hypoxia via intracellular adenylyl cyclase (AC) activation. Furthermore, we provide evidence for postsynaptic modulation of the hypoxic signal in sensory neurons originating via activation of presynaptic D$_2$R.

## Methods

### Ethical approval

All wild-type and transgenic zebrafish were bred and maintained at the Laboratory for the Physiology and Genetics of Aquatic Organisms, University of Ottawa. Zebrafish were maintained under a 14:10 h light/dark photocycle at 28°C (Westerfield, 2007). Embryos and larvae were reared on a diet of rotifers and Gemma 75 (Skretting Canada, S. Andrews, NB, Canada), juveniles were fed Artemia and Gemma 150–300 (Skretting) and adults were fed Adult Zebrafish Diet (Ziegler Feeds, Gardners, PA, USA). Adult zebrafish were killed by concussion and decapitated. All procedures for animal use and killing were carried out in accordance with institutional guidelines according to protocol BL-3666, and guidelines provided by the Canadian Council on Animal Care. The present work complies with the ethical guidelines set out by the journal and with those according to Animals in Research: Reporting In Vivo Experiments (ARRIVE).

In the present study, we used transgenic zebrafish Tg(*elavl3*:GCaMP6s) expressing the genetically encoded Ca$^{2+}$ indicator GCaMP6s under the pan-neuronal promotor *elavl3* (Dunn et al., 2016). GCaMP6s contains the green fluorescent protein (GFP) as part of its structure. The Tg(*dat:tom20 MLS-mCherry*) line has previously been used to visualize dopaminergic neurons (Reed et al., 2024). In this line, the regulatory elements of the dopamine transporter gene (*dat*) were targeted to a reporter, mCherry, after fusion with the mitochondrial localizing signal (MLS) of Tom20 (Noble et al., 2015). To determine the distribution of dopaminergic nerve terminals relative to GFP-positive NECs, adult Tg(*elavl3*:GCaMP6s) fish were crossed with adult Tg(*dat:tom20 MLS-mCherry*) to generate double transgenic offspring containing both mCherry and GFP.

### Immunohistochemistry

Techniques for tissue extraction and immunolabeling were carried out as previously described (Jonz & Nurse, 2003). Following termination by concussion and decapitation, whole gill baskets were removed and immersed in phosphate-buffered solution (PBS) containing (mM): 137 NaCl, 15.2 Na$_2$HPO$_4$, 2.7 KCl and 1.5 KH$_2$PO$_4$ at pH 7.8 (Bradford et al., 1994). Gill baskets were fixed by immersion in 4% paraformaldehyde in PBS overnight at 4°C. Tissues were removed and rinsed in PBS three times for 3 min before permeabilization for 24 h at 4°C. Permeabilizing solution (PBS-TX) contained 0.5–2% Triton X-100 in PBS (pH 7.8). After three rinses in PBS, gill baskets were then separated into individual arches. Gill arches were incubated in primary antibodies

for 24 h at 4°C, rinsed with PBS three times for 3 min and immersed in secondary antibodies for 1 h at room temperature in the dark.

NECs were identified using antibodies against 5-HT or synaptic vesicle protein SV2. Polyclonal anti-5-HT was raised in rabbit against a 5-HT creatinine sulphate complex conjugated with bovine serum albumin (manufacturer specifications; catalog. no. S5545; Sigma-Aldrich, Oakville, ON, Canada; Antibody Registry ID: AB_477522). Anti-5HT was used at a dilution of 1:250 and localized with goat anti-rabbit secondary antibodies conjugated with fluorescein isothiocyanate (dilution 1:50; catalog. no. 111-095-003; Cedarlane, Burlington, ON, Canada). Monoclonal SV2 raised in mouse (AB_2315387 and AB_531908; Developmental Studies Hybridoma Bank, University of Iowa, IA, USA) was used at a dilution of 1:100 and targeted by goat anti-mouse secondary antibodies conjugated with Alexa 594 at a dilution of 1:100 (catalog. no. A11005; Invitrogen, Burlington, ON, Canada). Neurons and nerve fibres were identified using antibodies against a zebrafish-specific neuronal marker (zn-12). Monoclonal anti-zn-12 raised in mouse (RRID: AB_2315387; Developmental Studies Hybridoma Bank, University of Iowa) was used at a dilution of 1:100 and visualized using goat anti-mouse secondary antibodies conjugated with Alexa 594 at a dilution of 1:100 (catalog. no. A11005; Invitrogen, Burlington, ON, Canada). Labelling by these antibodies in the zebrafish gill has been previously characterized (Jonz & Nurse, 2003).

### Relative $[Ca^{2+}]_i$ measurements

Following termination by concussion and decapitation, whole gill baskets were removed and separated into individual gill arches and immersed in extracellular solution containing (mm): 120 NaCl, 5 KCl, 2.5 CaCl$_2$, 2 MgCl$_2$, 10 Hepes and 10 glucose at pH 7.8. Isolated intact gill arches from Tg(*elavl3*:GCaMP6s) zebrafish were secured in a Petri dish using a metal tissue anchor (catalog. no. 640251; Warner Instruments, Holliston, MA, USA) and continuously perfused with extracellular solution at pH 7.8. GFP-fluorescing cells were observed using a 40× water-immersion objective (Nikon, Tokyo, Japan). NECs were identified by their size, presence of GFP, position along the centre of the filament epithelium and location at the distal end of the filament. Using a Lambda DG-5 wavelength changer (Sutter Instruments, Novato, CA, USA), the preparation was exposed to 490 nm excitation light for 600 ms at a sampling frequency of 1 s$^{-1}$. Images were captured with a CCD camera (QImaging, Surrey, BC, Canada) and fluorescence intensity was recorded with NIS Elements software (Nikon).

Dual recordings of NECs and ChNs were carried out as above, with a modified depth of focus. Because these two cell types are located in different layers of the gill filament, the recording plane was set at an intermediate depth between the NEC and ChN, where both cell types were slightly out of focus, but still within the field of view, and therefore could be recorded simultaneously.

### Solutions and drug treatments

High K$^+$ extracellular solution was prepared with 90 mm NaCl, 35 mm KCl, 2.5 mm CaCl$_2$, 2 mm MgCl$_2$, 10 mm Hepes and 10 mm glucose. Zero Ca$^{2+}$ extracellular solution was prepared with 120 mm NaCl, 5 mm KCl, 4.5 mm MgCl$_2$, 10 mm Hepes, 10 mm glucose and 1 mm EGTA. pH of all solutions was maintained at 7.8. 100% N$_2$ was bubbled through an air stone into solution reservoirs to create hypoxic solutions with a $P_{O_2}$ of ∼25 mmHg. All control solutions were bubbled with compressed air for the same duration of time.

Drugs were introduced into the recording chamber by perfusion in extracellular solution. Nifedipine (catalog. no. N7643; Sigma-Aldrich) and dantrolene (catalog. no. 0507; Tocris, Bristol, UK) were used to block entry of extracellular Ca$^{2+}$ and release of stored Ca$^{2+}$, respectively. To target dopamine receptors, the D$_2$R antagonist, domperidone (catalog. no. D122; Sigma-Aldrich) and the D$_2$R agonist, quinpirole (catalog. no. Q102; Sigma-Aldrich), were tested. To identify an intracellular mechanism for D$_2$R, the AC inhibitor SQ22536 (catalog. no. 1435; Tocris) and AC activator forskolin (catalog. no. 11018; Cayman Chemical, Ann Arbor, MI) were tested. All drugs were first dissolved in dimethyl sulphoxide (DMSO) to produce a final DMSO concentration of <0.1%. At this concentration, DMSO had no effect on Ca$^{2+}$ baseline and did not produce changes in fluorescence intensity (relative $[Ca^{2+}]_i$).

### Statistical analysis

For all reported data, sample size (*n*) refers to individual cells. Although multiple gill arches were assessed per animal, only one cell from each arch was included in the analysis to avoid repeated exposures or treatments in the same tissue. Throughout the study, hypoxic responses from a total of 73 cells were recorded from 41 adult zebrafish. For each recording, the baseline fluorescence was calculated as the average fluorescence intensity of a cell for the first 30 s in normoxia. All fluorescence values were divided by the baseline to evaluate changes in fluorescence intensity over time throughout a single recording. Statistical analysis for comparison of two groups was carried out using the Kruskal–Wallis test with Prism (GraphPad Software Inc., San Diego, CA, USA). All data are expressed as the mean ± SD.

## Results

### NECs contain GCaMP6s and display a Ca$^{2+}$ response to hypoxia *in situ*

The transgenic Tg(*elavl3*:GCaMP6s) zebrafish used in this study expresses a genetically-encoded Ca$^{2+}$ reporter, GCaMP, under the control of the pan-neuronal promotor, *elavl3* (Dunn et al., 2016). To confirm expression of GCaMP in chemoreceptors, we used immuno-histochemistry to identify GFP (part of the GCaMP complex) with known markers of gill NECs. Labelling of GFP-positive GCaMP-containing cells colocalized with anti-SV2 (Fig. 1*A–C*) and anti-5-HT (Fig. 1*D–F*), demonstrating that GCaMP is contained within NECs. We developed a procedure to record brief elevations in [Ca$^{2+}$]$_i$ that were associated with a hypoxic stimulus in single chemoreceptors *in situ* using isolated gills (Fig. 2). GCaMP-positive NECs were first identified using brightfield and 490 nm illumination (Fig. 2*B*) and then confirmed after successful Ca$^{2+}$-imaging experiments by fixation and immunohistochemistry (Fig. 2*C*). These GCaMP-positive NECs displayed a Ca$^{2+}$ response to hypoxia (Fig. 2*D*) and the magnitude of these responses did not change over time, nor after multiple exposures (Kruskal–Wallis test, $P > 0.999$, $n = 7$ cells) (Fig. 2*E*).

### Extracellular and stored Ca$^{2+}$ contribute to the NEC response to hypoxia

A combination of intracellular and extracellular blockers was used to evaluate the source of Ca$^{2+}$ in the NEC response to hypoxia. In experiments where NECs were exposed to successive bouts of hypoxia, the Ca$^{2+}$ response was significantly reduced by 50.4% with the addition of 100 μm nifedipine, an L-type Ca$^{2+}$ channel blocker (Fig. 3*A*). The Ca$^{2+}$ response fully recovered after a 15 min washout of nifedipine (Kruskal–Wallis test, $P = 0.006$, $n = 6$ cells) (Fig. 3*A* and *B*). Similarly, the response to hypoxia was partially reduced by 41.9% when Ca$^{2+}$ was removed from the extracellular solution (Kruskal–Wallis test, $P < 0.0059$, $n = 6$ cells) (Fig. 3*C* and *D*).

The NEC Ca$^{2+}$ response to hypoxia was partially reduced by 30.4% with the addition of dantrolene, an inhibitor of intracellular Ca$^{2+}$ release (Kruskal–Wallis test, $P = 0.007$, $n = 6$ cells) (Fig. 4*A* and *C*). To confirm that the Ca$^{2+}$ response to hypoxia was a reflection of the sum of Ca$^{2+}$ arising from both intracellular and extracellular

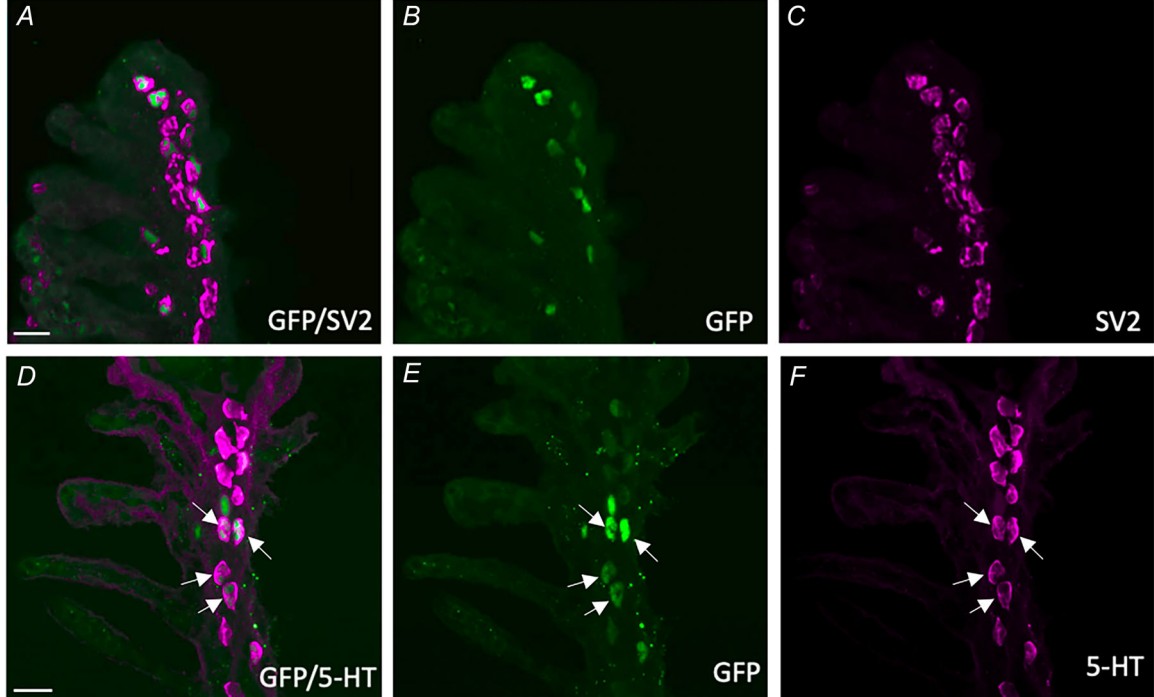

**Figure 1. Characterization of GCaMP-positive neuroepithelial cells (NECs) in the gill epithelium of transgenic *elavl3*:GCaMP6s zebrafish**
Confocal imaging of immunohistochemical localization of GCaMP with NECs containing synaptic vesicle protein-2 (SV2) and 5-hydroxytryptamine (5-HT). *A*, labelling with GFP (green) co-localized with NECs containing SV2 (magenta). *B* and *C*, GCaMP and SV2 labelling shown separately. Scale bar = 20 μm in (*A*) to (*C*). *D*, labelling with GCaMP (green) co-localized with NECs containing 5-HT (magenta, arrows). *E* and *F*, GCaMP and 5-HT labelling shown separately. Scale bar = 20 in (*D*) to (*F*). [Colour figure can be viewed at wileyonlinelibrary.com]

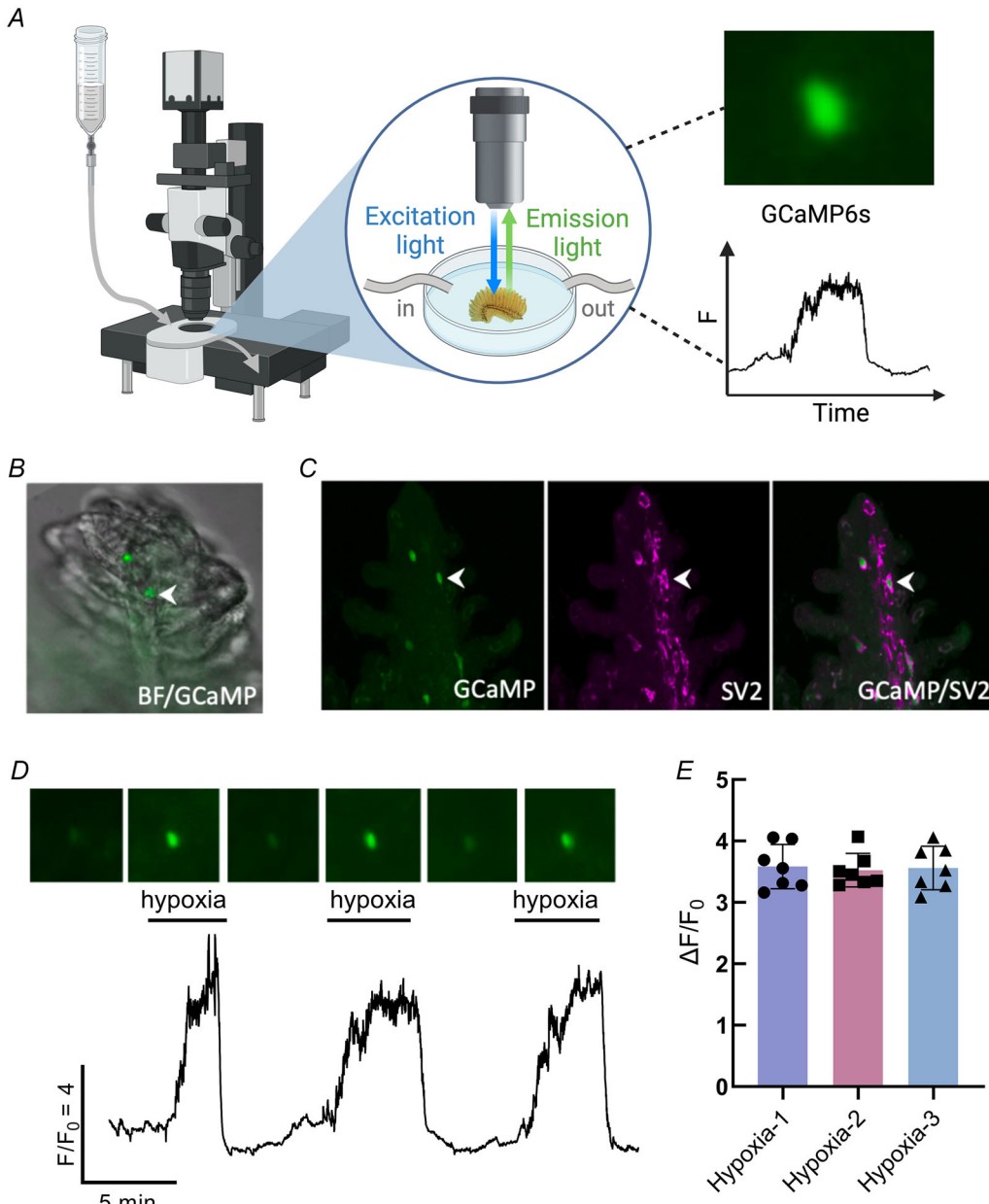

**Figure 2. Hypoxia induced intracellular Ca²⁺ responses in gill neuroepithelial cells from Tg(*elavl3*:GCaMP6s) zebrafish**

*A*, schematic of the GCaMP recording preparation illustrating a fluorescence microscope (left), an isolated gill in a recording chamber fitted with in- and out-flow for superfusion (centre) and fluorescence excitation in a cell during a hypoxic stimulus (right). *B*, overlay of brightfield and green fluorescence (488 nm) images of a GCaMP-positive neuroepithelial cell (NEC, arrowhead) *in situ* containing GCaMP from a Tg(*elavl3*:GCaMP6s) zebrafish. *C*, *post hoc* confocal imaging confirming immunohistochemical co-localization of GCaMP (green) with synaptic vesicle protein-2 (SV2, magenta) in the NEC (arrowheads) identified in (*B*). Left: GCaMP and SV2 separately, and GCaMP and SV2 labelling together. *D*, Ca²⁺ imaging trace from the GCaMP-containing cell in (*A*) during three bouts of hypoxia. Scale indicates time (min) and relative changes in fluorescence (*F/F₀*) corresponding to changes in intracellular Ca²⁺ concentration ([Ca²⁺]ᵢ). Time-series micrographs above show fluorescence changes over time. *E*, mean ± SD *F/F₀* in NECs in response to three consecutive bouts of hypoxia. There was no significant change in the magnitude of the Ca²⁺ response to hypoxia over time (Kruskal–Wallis test, *P* > 0.999, *n* = 7 cells). The schematic in (*A*) was created with BioRender.com. [Colour figure can be viewed at wileyonlinelibrary.com]

sources, we exposed NECs to hypoxia in the presence of dantrolene and $Ca^{2+}$-free solution (Kruskal–Wallis test, $P < 0.0079$, $n = 5$ cells) (Fig. 4*B* and *D*). Compared with all other treatments, blocking both intracellular and extracellular $Ca^{2+}$ resulted in the largest reduction in the $Ca^{2+}$ response to hypoxia (Fig. 4*E*).

### The NEC $Ca^{2+}$ response to hypoxia is reduced by $D_2R$ activity

Previous studies reported a decrease in ventilation frequency of larval zebrafish with dopamine or the specific dopamine $D_2R$ agonist, quinpirole (Reed et al., 2024; Shakarchi et al., 2013). Additionally, immunohistochemical labelling has shown co-localization of dopamine $D_2R$ with NECs and has provided evidence for the postsynaptic synthesis and reuptake of dopamine in the gill (Reed et al., 2024). The present study aimed to determine an inhibitory role for $D_2R$ within the gill at the level of the presynaptic NEC. We found the NEC $Ca^{2+}$ response to hypoxia was reversibly reduced by 44.1% with the addition of dopamine (Kruskal–Wallis test, $P = 0.0116$, $n = 5$ cells) (Fig. 5*A* and *B*), as well as reduced by 57.9% with the addition of quinpirole (Kruskal–Wallis

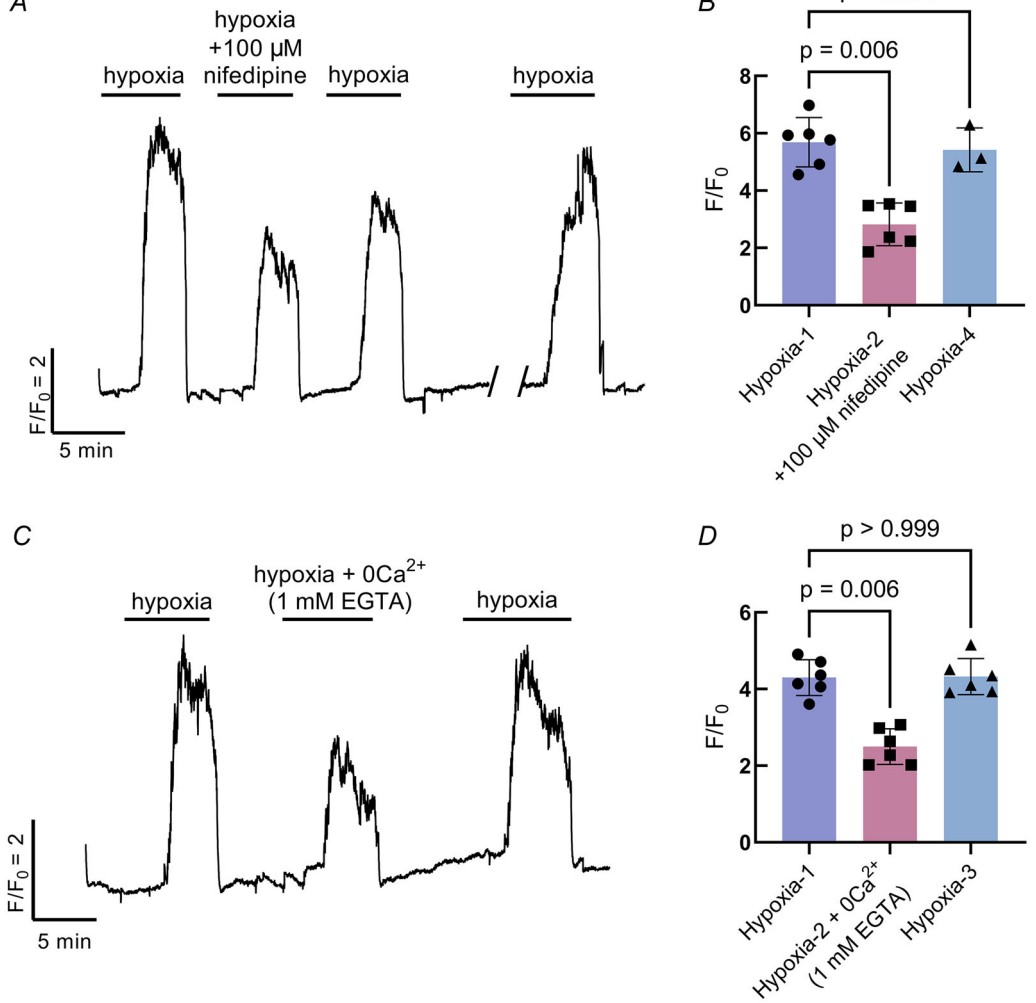

**Figure 3. Extracellular $Ca^{2+}$ contributes to the response to hypoxia in neuroepithelial cells (NECs)**
*A*, $Ca^{2+}$ imaging trace from a GCaMP-containing NEC, where the response to hypoxia was reduced with the addition of 100 μM nifedipine, an L-type $Ca^{2+}$ channel blocker. The response to hypoxia was fully recovered after a 15 min washout (break in trace). *B*, Summary data from NECs treated as in (*A*) illustrating a reduction in the mean ± SD *F/F$_0$* (Kruskal–Wallis test, $P = 0.006$, $n = 6$ cells). Three cells were evaluated for recovery after a 15 min washout period (Kruskal–Wallis test, $P > 0.999$, $n = 3$ cells). *C*, $Ca^{2+}$ imaging trace from a GCaMP-containing NEC, where the response to hypoxia was reversibly reduced when $Ca^{2+}$ was removed from the extracellular solution. *D*, summary data from NECs treated as in (*C*) showing a reduction in the mean ± SD *F/F$_0$* (Kruskal–Wallis test, $P < 0.006$, $n = 6$ cells). The response to hypoxia fully recovered after zero $Ca^{2+}$ treatment (Kruskal–Wallis test, $P > 0.999$). [Colour figure can be viewed at wileyonlinelibrary.com]

test, $P = 0.007$, $n = 6$ cells) (Fig. 5*C* and *D*). Domperidone, a D$_2$R antagonist, had the opposite effect and enhanced the NEC Ca$^{2+}$ response to hypoxia by 26.7% compared to hypoxia alone (Kruskal–Wallis test, $P = 0.0347$, $n = 6$ cells) (Fig. 5*E* and *F*).

To evaluate an intracellular pathway for the inhibition of the NEC Ca$^{2+}$ response to hypoxia via dopamine D$_2$Rs, we used drugs to target AC. Activation of D$_2$R reduces AC activity (Usiello et al., 2000). We found a decrease in the response to hypoxia with the addition of SQ22536, an inhibitor of AC (Kruskal–Wallis test, $P = 0.0099$, $n = 6$ cells) (Fig. 6*A* and *B*). Furthermore, when the Ca$^{2+}$ response to hypoxia was reduced in the presence of dopamine, activation of AC with forskolin partially restored the normal Ca$^{2+}$ response to hypoxia (Kruskal–Wallis test, $P = 0.0019$, $n = 6$ cells) (Fig. 6*C* and *D*).

## Postsynaptic responses to hypoxia are modulated by presynaptic D$_2$R activity

In addition to NECs, immunohistochemical labelling confirmed that SV2/zn-12-positive ChNs in the gill of Tg(*elavl3*:GCaMP6s) zebrafish are GCaMP-positive (Fig. 7). ChNs extend the complete length of gill filaments

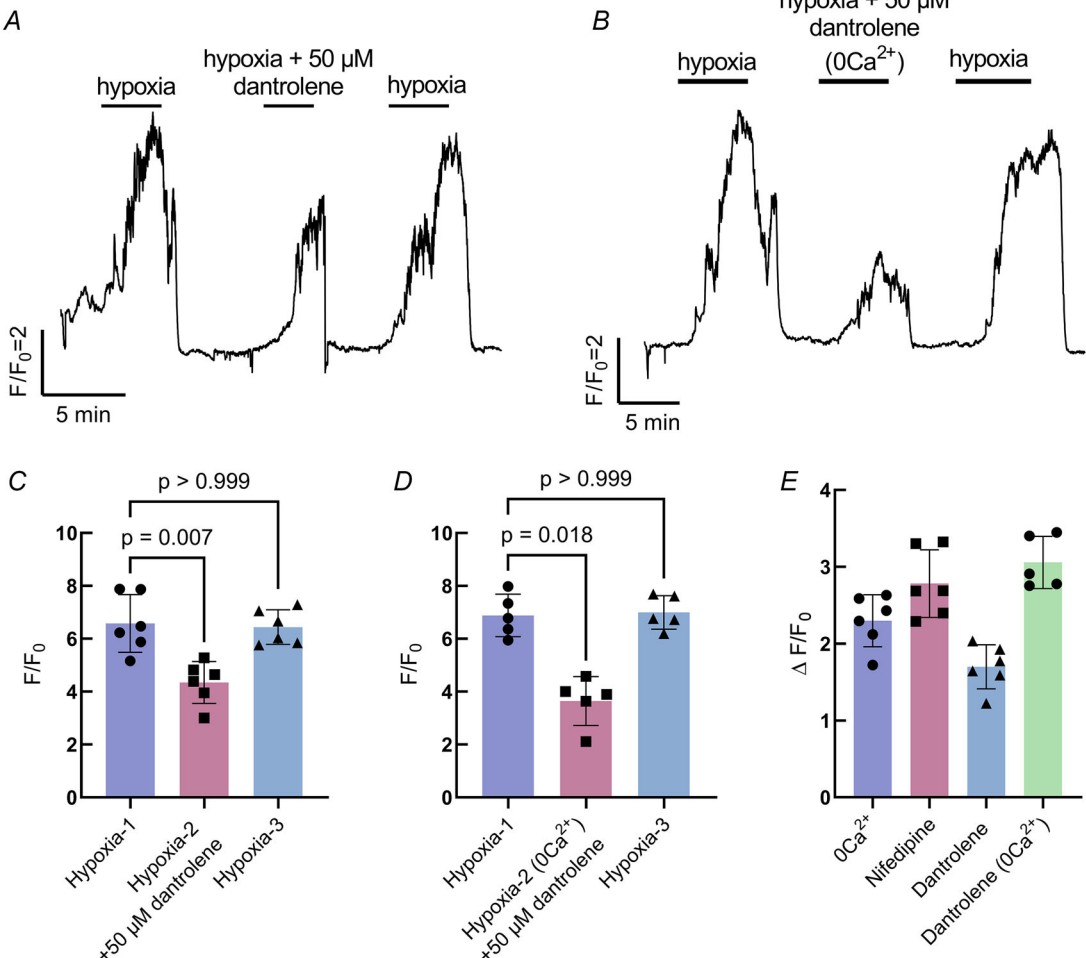

**Figure 4. Intracellular Ca$^{2+}$ contributes to the response to hypoxia in neuroepithelial cells (NECs)**
*A*, Ca$^{2+}$ imaging trace from a GCaMP-containing NEC, where the response to hypoxia was reversibly reduced with the addition of 50 μM dantrolene, an inhibitor of intracellular Ca$^{2+}$ release. *B*, Ca$^{2+}$ imaging trace from a GCaMP-containing NEC demonstrating the combined contributions of intracellular and extracellular Ca$^{2+}$. The response to hypoxia was further reduced with the addition of 50 μM dantrolene in Ca$^{2+}$-free extracellular solution. *C*, summary data treated as in (*A*) showing reduction in the mean $\pm$ SD $F/F_0$ (Kruskal–Wallis test, $P = 0.007$, $n = 6$ cells). The response to hypoxia fully recovered (Kruskal–Wallis test, $P > 0.999$, $n = 6$ cells). *D*, summary data treated as in (*B*) showing a reduction in the mean $\pm$ SD $F/F_0$ (Kruskal–Wallis test, $P < 0.008$, $n = 5$ cells). The response to hypoxia fully recovered (Kruskal–Wallis test, $P > 0.999$, $n = 5$ cells). *E*, summary comparing all Ca$^{2+}$ blocking treatments. The combination of dantrolene with Ca$^{2+}$-free extracellular solution resulted in an additive effect compared to dantrolene alone. [Colour figure can be viewed at wileyonlinelibrary.com]

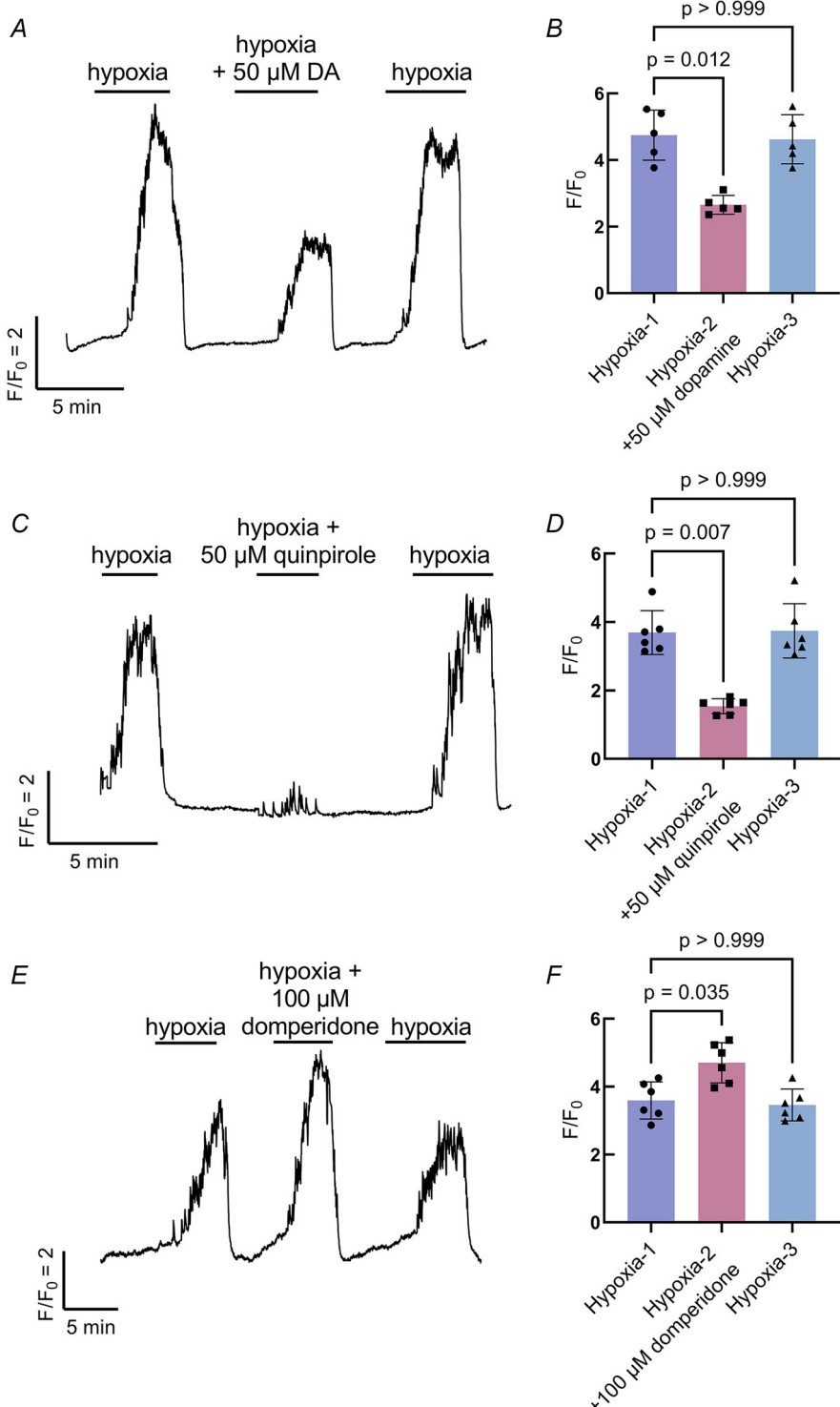

**Figure 5. The effects of D$_2$R activity on the neuroepithelial cell (NEC) response to hypoxia**

Showing the effects of dopamine (*A* and *B*), quinpirole (*C* and *D*) and domperidone (*E* and *F*). *A*, Ca$^{2+}$ imaging trace from a GCaMP-containing NEC where the response to hypoxia was reversibly reduced with the addition of 50 μM dopamine (DA). *C*, Ca$^{2+}$ imaging trace from a GCaMP-containing NEC where the response to hypoxia was reversibly reduced with the addition of 50 μM quinpirole, a specific dopamine D$_2$R agonist. *E*, Ca$^{2+}$ imaging trace from a GCaMP-containing NEC where the response to hypoxia was enhanced with the addition of 100 μM domperidone, a specific dopamine D$_2$R antagonist. *B*, *D* and *F*, summary data showing the mean ± SD. *F/F$_0$* corresponding to experiments in (*A*), (*C*) and (*E*). Addition of 50 μM dopamine significantly reduced the Ca$^{2+}$

response to hypoxia (Kruskal–Wallis test, $P = 0.012$, $n = 5$ cells) (*B*), as well as 50 μM quinpirole (Kruskal–Wallis test, $P = 0.007$, $n = 6$ cells) (*D*), wheras domperidone increased the mean ± SD $F/F_0$ (Kruskal–Wallis test, $P = 0.035$, $n = 6$ cells) (*F*). The response to hypoxia fully recovered following all treatments (Kruskal–Wallis test, $P > 0.999$, $n = 5$–6 cells). [Colour figure can be viewed at wileyonlinelibrary.com]

and were previously shown to innervate NECs in zebrafish (Jonz & Nurse, 2003). Rotation of confocal images by 90° showed zn-12-positive nerve fibres in close association with GCaMP-positive NECs and ChNs (Fig. 7*D*–*I*).

In our whole-gill recording preparation, multiple ChNs of a single gill filament showed nearly simultaneous $Ca^{2+}$ responses to hypoxia (Fig. 8*A* and *B*). Because NECs are confined to the distal end of the gill filament, the filament was transected at a proximal region where ChNs were present but no NECs were observable (Fig. 8*C*). This technique was used to mechanically remove synaptic

contact between NECs and ChNs to determine whether ChNs can respond to hypoxia independently of NECs. After the filament was cut, the ChN response to hypoxia was completely abolished, although ChNs were still able to show a response to high extracellular $K^+$, indicating that ChNs were not adversely affected by the cut (Fig. 8*D*). In these experiments, we observed a mean $F/F_0$ response of $4.84 \pm 0.67$ ($n = 5$) to hypoxia before nerve transection, compared to $5.46 \pm 1.03$ ($n = 5$) produced by high $K^+$.

To further confirm that NECs were innervated by ChNs, fish were treated with 6-OHDA, a neurotoxin

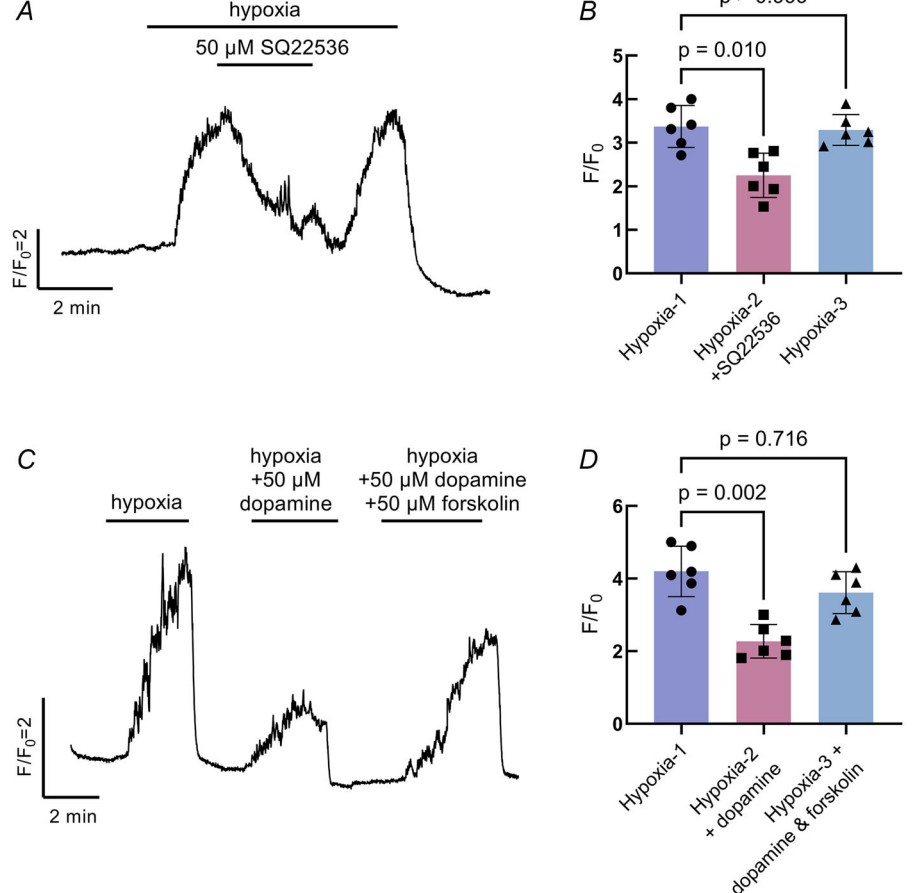

**Figure 6. Dopamine acts through intracellular secondary messenger cAMP in neuroepithelial cells (NECs)**
*A*, addition of SQ22536, an adenylyl cyclase (AC) inhibitor, decreased the effect of hypoxia on intracellular $Ca^{2+}$. *B*, summary data treated as in (*A*) showing the mean ± SD $F/F_0$ during the first 2 min of hypoxia exposure (Hypoxia-1), the reduction in the mean ± SD $F/F_0$ during the entire duration of SQ22536 exposure (Hypoxia-2 + SQ22536; Kruskal–Wallis test, $P = 0.010$, $n = 6$ cells) and the mean ± SD $F/F_0$ during the last 2 min of hypoxia exposure (Hypoxia-3). *C*, forskolin, an AC activator, partially recovered the suppressive effect of dopamine on the $Ca^{2+}$ response to hypoxia. *D*, summary data treated as in (*C*) showing recovery in the mean ± SD $F/F_0$ of the hypoxic response from dopamine with forskolin (Kruskal–Wallis test, $P = 0.002$, $n = 6$ cells). The response to hypoxia fully recovered following both treatments (Kruskal–Wallis test, $P > 0.999$, $n = 6$ cells). [Colour figure can be viewed at wileyonlinelibrary.com]

used to destroy the nerve terminals of dopaminergic neurons. Because NECs are innervated by nerve fibres containing the dopamine active transporter (DAT) (Reed et al., 2024), this technique was used to chemically remove the ability of postsynaptic ChNs to receive inputs from NECs. In transgenic animals produced by crossing the Tg(*dat:tom20 MLS-mCherry*) and Tg(*elavl3*:GCaMP6s) lines, labelling of DAT-positive ChN terminals around NECs was partially reduced in fish treated with 6-OHDA (Fig. 8*E*). Importantly, after treatment with 6-OHDA, ChNs showed no response to hypoxia at the same time as maintaining the ability to respond to high K$^+$ (overlaid traces, $n = 6$ cells) (Fig. 8*F*).

Finally, we aimed to link our characterization of D$_2$R activity in NECs with the Ca$^{2+}$ response in postsynaptic ChNs. High K$^+$ was used to depolarize the membranes of ChNs with and without the addition of quinpirole. In

agreement with the reported absence of D$_2$R on ChNs (Reed et al., 2024), both 50 μM and 100 μM quinpirole failed to reduce the [Ca$^{2+}$]$_i$ response during a high K$^+$ stimulus, confirming that dopamine and quinpirole do not affect ChNs directly and must be acting pre-synaptically (Fig. 9*A* and *B*). In dual recording of NECs and ChNs, in which synaptic contact between cells was left intact, [Ca$^{2+}$]$_i$ was simultaneously recorded in both cell types during exposure to hypoxia (Fig. 9*C*). Under these conditions, NECs responded first with an increase in [Ca$^{2+}$]$_i$, followed by a [Ca$^{2+}$]$_i$ response in ChNs after a latency of 25.6 ± 4.7 s ($n = 5$) (Fig. 9*D*). There was a reduction in both the NEC and ChN Ca$^{2+}$ response to hypoxia with the D$_2$R agonist, quinpirole (NEC: Kruskal–Wallis test, $P = 0.003$, $n = 5$ cells, and ChN: Kruskal–Wallis test, $P = 0.013$, $n = 5$ cells) (Fig. 9*D* and *E*).

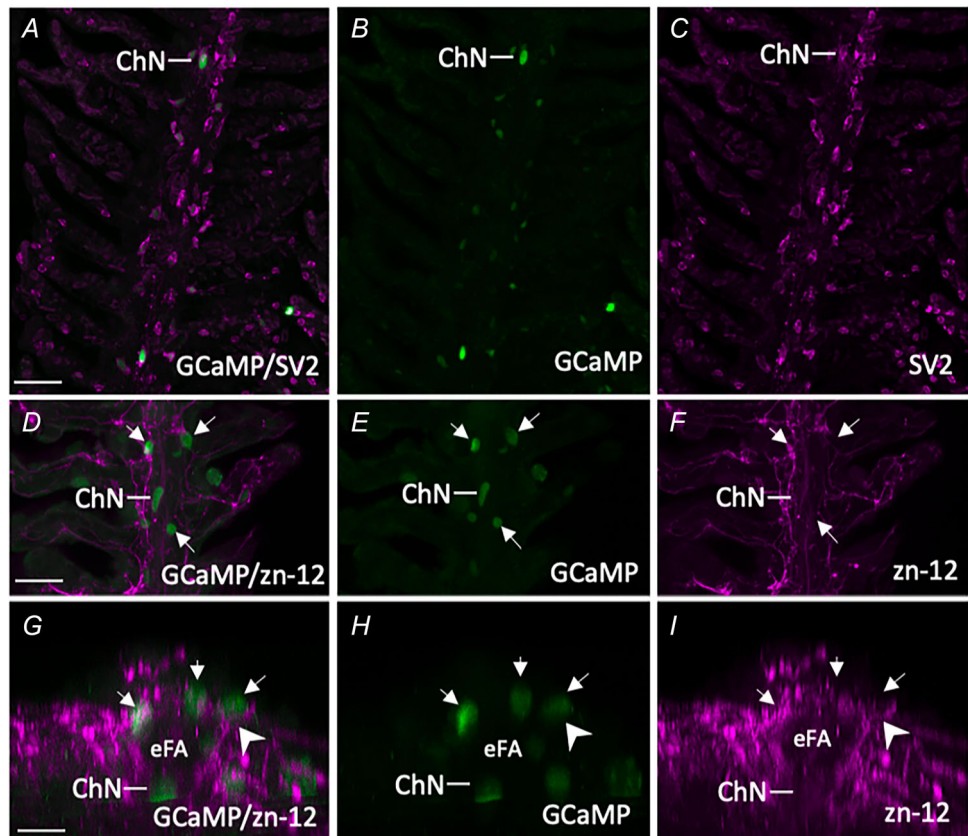

**Figure 7. Characterization of GCaMP-positive postsynaptic chain neurons (ChNs) in Tg(*elavl3*:GCaMP6s) zebrafish gills**

*A–C*, confocal imaging of immunohistochemical localization of GCaMP in a ChN containing synaptic vesicle protein-2 (SV2, magenta). *B* and *C*, GCaMP and SV2 labelling shown separately. Scale bar = 50 μm in (*A*) to (*C*). *D–F*, co-labelling of GCaMP-positive ChNs and NECs (green, arrows) with nerve fibres labelled with zn-12 (magenta). *E* and *F*, GCaMP and zn-12 labelling shown separately. Scale bar = 50 μm in (*D*) to (*F*). *G–I*, images from (*D*) to (*F*), cropped and titled back 90°. Rotation demonstrates neural connection (arrowhead) between nerve fibres located below the efferent filament artery (eFA) where ChNs are located, projecting to GCaMP-positive NECs. [Colour figure can be viewed at wileyonlinelibrary.com]

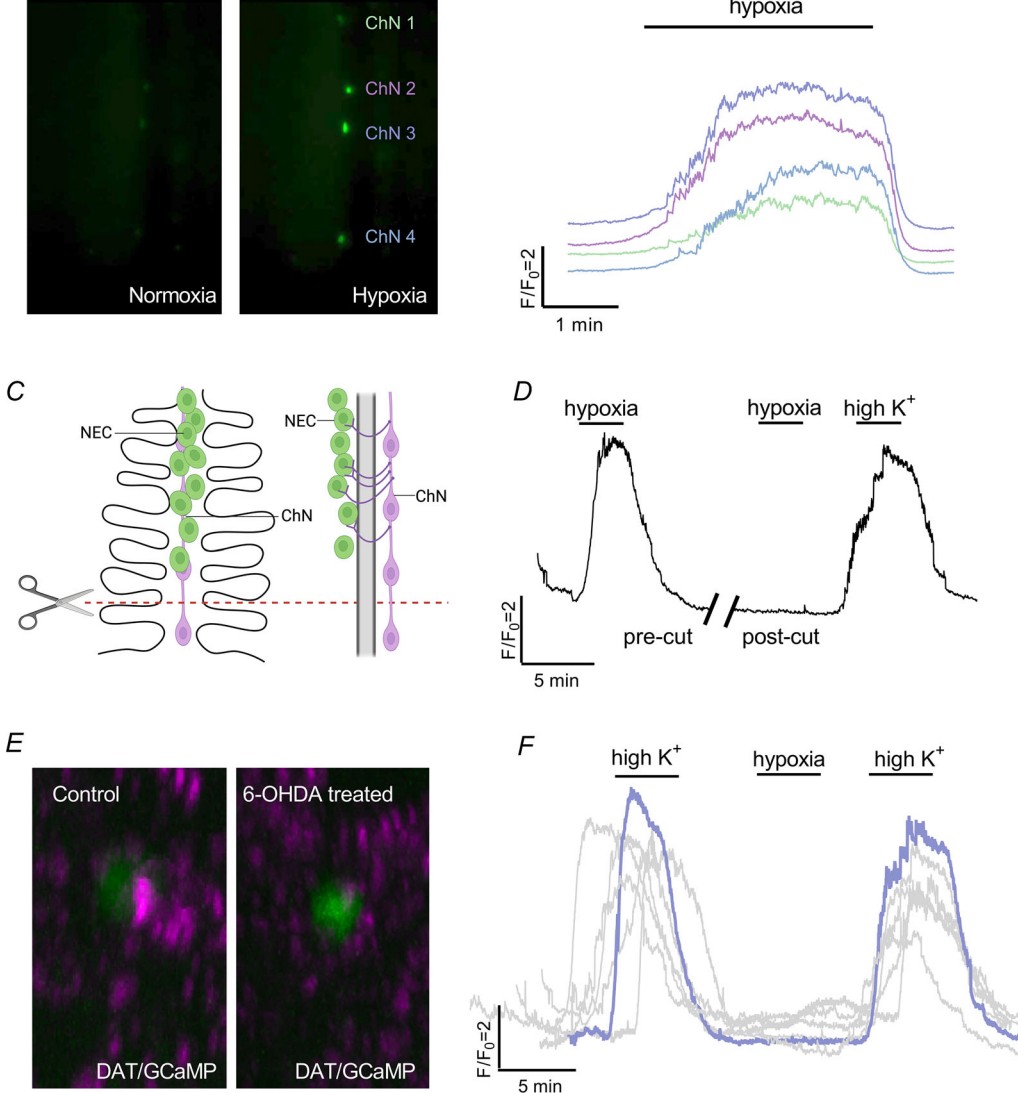

**Figure 8. The chain neuron (ChN) calcium response to hypoxia requires synaptic contact with neuro-epithelial cells (NECs)**

*A*, examples from live fluorescence imaging at 488 nm of four ChNs along a single filament in normoxia (left micrograph) and hypoxia (right micrograph). *B*, calcium traces from the four ChNs shown in (*A*) responding to hypoxia. Different colours represent corresponding cells (ChN 1–4) in (*A*). *C*, schematic of a gill filament (left) and rotation by 90° on the *y*-axis (right) illustrate filament transection. Filaments were cut along the red dashed line at the proximal end where a ChN was present, but no NECs were observable. *D*, calcium imaging trace of a single ChN before and after filament transection. After the filament was cut (break in trace), the neuron no longer responded to hypoxia (*n* = 5 cells). As a positive control, viability of the neuron was demonstrated by stimulation with a solution of high extracellular K⁺. *E*, confocal imaging of gills from a double transgenic animal produced by crossing Tg(*elavl3*:GCaMP6s) and Tg(*dat:tom20 MLS-mCherry*) fish showing the relationship between the dopamine active transporter (DAT) nerve endings (magenta) and GCaMP-positive NECs (green) in control (left micrograph) and 6-OHDA-treated gills (right micrograph). DAT labelling was found in close proximity to NECs but was reduced after 6-OHDA treatment. *F*, overlayed Ca²⁺ imaging traces from 6-OHDA treated animals. After 6-OHDA treatment, the ChNs did not respond to hypoxia (*n* = 6 cells). Response to high extracellular K⁺ confirmed cell viability. The schematic in (*C*) was created with BioRender.com. [Colour figure can be viewed at wileyonlinelibrary.com]

# Discussion

The present study demonstrates neuromodulation of the chemoreceptor response to hypoxia produced by activation of $D_2R$ in NECs in the gills of zebrafish. We used a transgenic zebrafish line in which the $Ca^{2+}$ reporter, GCaMP, was genetically encoded to reveal dopaminergic inhibition of the NEC $Ca^{2+}$ response to hypoxia, acting through $D_2R$ and inhibition of AC. We further demonstrated a link between the inhibitory effect of activation of presynaptic $D_2R$ and modulation of the hypoxic signal in postsynaptic neurons that innervate NECs.

## The gill as a model for the chemoreceptor response to hypoxia

Evaluating responses of chemoreceptors to hypoxia in previous work has typically relied on the techniques of classical $Ca^{2+}$ imaging, where the loading of $Ca^{2+}$ reporting dyes into isolated cells was required in the carotid body (Montoro et al. 1996), neuroepithelial bodies (Livermore et al., 2015) and gill NECs (Abdallah et al., 2015; Leonard et al., 2022; Porteus et al., 2014; Zhang et al., 2011). Here, we developed an intact, whole-gill preparation using the endogenous $Ca^{2+}$ indicator, GCaMP. This preparation allows us to observe chemoreceptor responses to hypoxia without the added complication of dye loading, at the same time as maintaining functional connectivity throughout the organ. Gill GCaMP-positive chemoreceptors showed a reliable response to hypoxia that was consistent over time and multiple hypoxic stimuli, serving as a useful model to evaluate chemoreceptor responses to hypoxia.

Although several studies have used $Ca^{2+}$ imaging to evaluate chemoreceptor activity in gill NECs (Abdallah et al., 2015; Leonard et al., 2022; Porteus et al., 2014; Zhang et al., 2011), the source of $Ca^{2+}$ contributing to

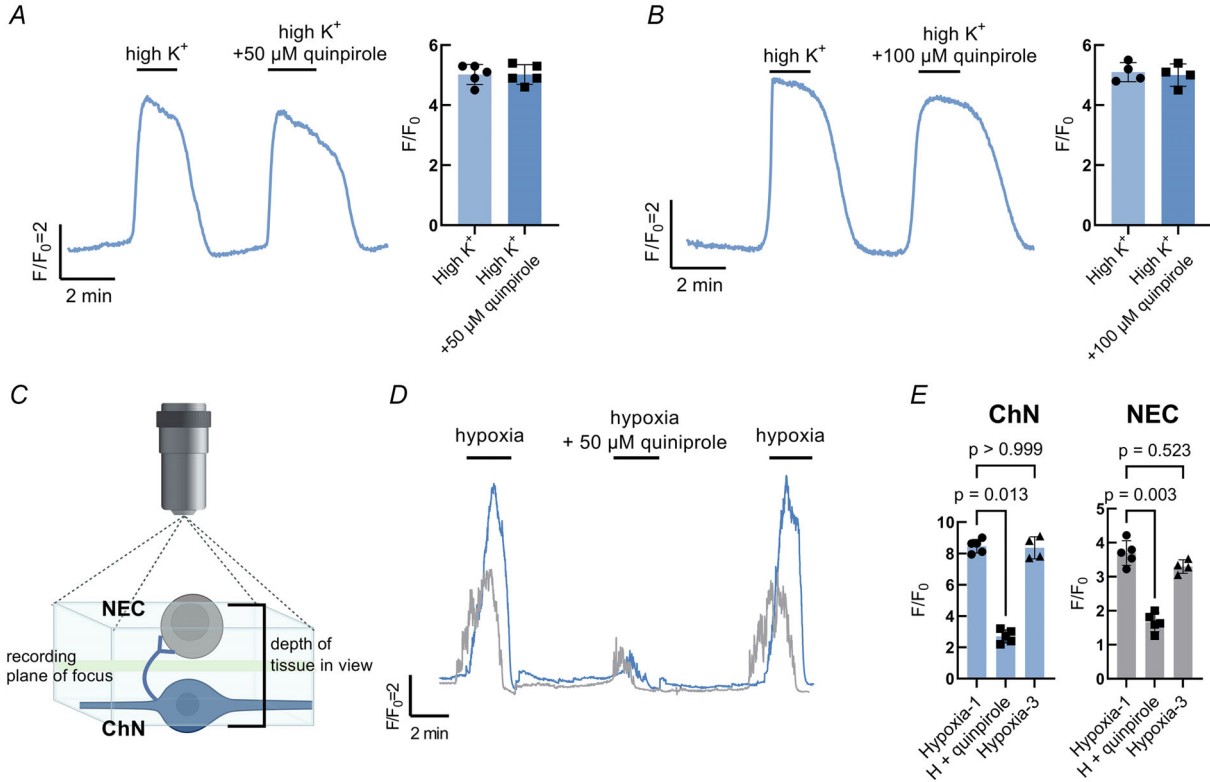

**Figure 9. Postsynaptic modulation of the hypoxic response by presynaptic D₂R activation**
*A*, calcium imaging trace from a single chain neuron with summary data showing no change in the response to high K⁺ with the addition of 50 μM quinpirole. *B*, calcium imaging trace from a single chain neuron (ChN) with summary data showing no change in the response to high K⁺ with the addition of 100 μM quinpirole. *C*, schematic illustration of preparation for dual recording of neuroepithelial cell (NEC) and ChN. Focus plane for recording was set at a tissue depth between both cells (green line) where both cells were still in view (blue box). *D*, dual-recording Ca²⁺ imaging trace of a NEC (grey) and ChN (blue) recorded simultaneously. *E*, summary data treated as in (*D*) showing a decrease in the hypoxic signal produced by ChNs (Kruskal–Wallis test, *P* = 0.013, *n* = 5 cells) and NECs (Kruskal–Wallis test, *P* = 0.003, *n* = 5 cells) with quinpirole. In both ChNs and NECs, the hypoxic response was fully recovered after quinpirole treatment (*P* > 0.999 or *P* = 0.523, *n* = 4 cells). The schematic in (*C*) was created with BioRender.com. [Colour figure can be viewed at wileyonlinelibrary.com]

the response to hypoxia in NECs had not clearly been determined. In the present study, we determined that the NEC $Ca^{2+}$ response to hypoxia involves both extracellular $Ca^{2+}$ influx through L-type $Ca^{2+}$ channels, as well as release from intracellular stores. Similarly in the carotid body, a combination of $Ca^{2+}$ entry through voltage-gated $Ca^{2+}$ channels and intracellular $Ca^{2+}$ release may be important in chemoreceptor responses to hypoxia (Kim et al., 2020).

## Dopaminergic modulation of chemoreceptor activity during hypoxia

In the present study, we demonstrated a reduction in the NEC $Ca^{2+}$ response to hypoxia produced by application of dopamine or the specific $D_2R$ agonist, quinpirole. $D_2Rs$ were previously localized to gill NECs in zebrafish by immunohistochemistry, whereas $D_2R$ was not found on postsynaptic ChNs (Reed et al., 2024). In addition, $D_2R$ expression in NECs was demonstrated using single-cell RNA-sequencing (Pan et al., 2022). Accordingly, the present results confirm that ChNs showed no direct response to quinpirole. We therefore attribute the suppressive effects of dopamine and quinpirole to activation of presynaptic $D_2Rs$. Dopamine receptors are a type of G-protein-coupled receptor playing a crucial role in mediating the effects of the neurotransmitter, dopamine. There are five known subtypes of dopamine receptors classified into two main families based on their signalling mechanisms: $D_1$-like ($D_1$ and $D_5$) and $D_2$-like ($D_2$, $D_3$ and $D_4$). Upon dopamine binding, dopamine receptors undergo conformational changes that activate intracellular signalling pathways through interactions with G-proteins. Activation of $D_2$-like receptors inhibits AC activity through $G_{\alpha i/o}$ proteins, leading to decreased cAMP levels within the cell (Beaulieu & Gainetdinov, 2011). We showed that inhibition of AC with SQ22536 mimicked the modulation of the NEC hypoxic response by dopamine. Furthermore, the modulatory effects of dopamine were reversed by addition of forskolin, an AC activator. Together, these results provide evidence for the dopaminergic inhibition of AC, and suggest subsequent decreases in cAMP activity, leading to modulation of the NEC $Ca^{2+}$ response to hypoxia. This mechanism of inhibition by $D_2R$ is similar to carotid body type 1 cells, where dopamine causes a dose-dependent decrease in the type 1 cell cAMP content induced by forskolin (Batuca et al., 2003). Downstream effectors of AC-cAMP on chemoreceptor activity may involve the modulation of $Ca^{2+}$ influx. For example, in endocrine cells from the rat, $D_2R$ activation has been associated with activation of $K^+$ channel currents, causing hyperpolarization, and the reduction of voltage gated $Ca^{2+}$ currents (Einhorn et al., 1991).

## Dopaminergic modulation of gill hypoxia signalling

The present study demonstrates modulation of the ChN response to hypoxia via presynaptic $D_2R$ activation, which suggests that $D_2R$ at the NEC-ChN synapse may act on ventilation. ChNs are part of a nerve bundle in the gill located beneath the efferent filament artery that sends projections to innervate NECs. Using gill transection experiments, we identified a pathway for the transmission of the hypoxic signal in the gill through a functional unit containing two parts: the postsynaptic ChN and any number of NECs innervated by that ChN. We further showed that, within this functional unit, the inhibitory actions of dopamine on the presynaptic NEC can be observed in the postsynaptic ChN, suggesting a mechanism of modulation at the NEC-ChN synapse. Moreover, in dual recordings, the $Ca^{2+}$ response to hypoxia in NECs always preceded the response in ChNs, suggesting that the hypoxic signal is carried from NECs to sensory ChNs. This mechanism may be similar to the carotid body, where dopamine released by type 1 cells has an autocrine–paracrine action on dopaminergic $D_2Rs$ located on type 1 cells to inhibit $Ca^{2+}$-channels, leading to negative feedback regulation of further neurotransmitter release during hypoxia (Benot & López-Barneo, 1990). The excitatory neurotransmitter at the NEC-ChN synapse has yet to be elucidated. Candidates include ACh, which is excitatory in the carotid body and is found in the zebrafish gill (Zachar et al., 2017b), and 5-HT, which is present in most NECs in zebrafish (Jonz & Nurse, 2003).

ChNs were previously shown to innervate NECs in gills of zebrafish (Jonz & Nurse, 2003), but their physiological role has been elusive. We propose that ChNs may act as interneurons that carry a hypoxic signal from NECs to afferent fibres of the glossopharyngeal or vagus nerves, comprising the cranial nerves that carry chemoreceptor activity to the CNS in fish (Burleson & Milsom, 1995; Sundin & Nilsson, 2002). ChNs contain varicose processes that indicate multiple locations of synaptic contact, possibly with the adjacent sensory fibres of the extrinsic nerve bundle (Jonz & Nurse, 2003). Furthermore, in zebrafish larvae, exogenous application of dopamine reduces ventilation frequency, whereas the $D_2R$ agonist, quinpirole, attenuates the hyperventilatory response to hypoxia (Reed et al., 2024; Shakarhi et al., 2013). These results, combined with the present findings, suggest that the observed decrease in hyperventilation is the result of presynaptic $D_2R$ activation and a reduction in NEC activity.

Intriguingly, in the present study, we report that the $D_2R$ antagonist, domperidone, enhanced the NEC $Ca^{2+}$ response to hypoxia. This observation points to endogenous dopamine release during hypoxia acting upon presynaptic $D_2Rs$ to limit the cellular response to hypoxia. Previous work has identified postsynaptic neurons,

including ChNs, as a source of dopamine production and storage in the gill (Reed et al., 2024). Presynaptic type 1 cells are largely responsible for synthesizing and releasing dopamine in the mammalian carotid body. However, many carotid body afferents are also dopaminergic (Finley et al., 1992), and carotid sinus nerve fibres innervating type 1 cells release dopamine (Almaraz et al., 1993), suggesting an additional postsynaptic source of dopamine to further modulate the type 1 cell response to hypoxia, similar to that observed in the gill.

## Implications

The results of the present study have implications for the field of oxygen sensing, and may contribute to our understanding of acclimatization, development and evolution in vertebrates. For example, modulation of presynaptic chemoreceptor inhibition by $D_2R$ may contribute to hypoxia acclimatization. In zebrafish, total gill $D_2R$ mRNA expression decreases after 48 h of hypoxia exposure, a timepoint in acclimation where there is a shift away from aquatic surface respiration behaviour, allowing the animal to rely more heavily on hyperventilation (Reed et al., 2024). Acclimation may lower the number of inhibitory $D_2Rs$ expressed by NECs and lead to a change in chemoreceptor sensitivity of the cell to dopamine. A similar involvement of $D_2Rs$ in hypoxia acclimation has been proposed in the carotid body by Huey and Powell (2000), who showed decreased carotid body $D_2R$ mRNA expression following hypoxia acclimation and suggested that this may be one mechanism by which exposure to chronic hypoxia reduces inhibition of hypoxia signalling to enhance ventilatory acclimatization to hypoxia.

The zebrafish model used in the present study may be particularly useful in characterizing the mechanisms involved in age-related changes in chemosensitivity. In the mammalian carotid body, the expression of genes encoding important components of the dopaminergic system is altered during development. These changes coincide with changes in chemoreceptor activity and output observed in early postnatal development in rats (Bairam & Carroll 2005; Gauda, 2002; Gauda et al., 1996). Zebrafish have already shown an adaptable dopaminergic system under hypoxia (Reed et al., 2024) and thus may serve as a great model for exploring changes in chemoreceptor sensitivity during development.

$D_2R$ is the predominant dopamine receptor in the carotid body, which is consistent with what we have found here in the zebrafish gill; however, that does not rule out the contributions of other dopamine receptors such as $D_1$. mRNA expression of $D_1$ receptors has been found in the carotid body of rats, cats and rabbits (Bairam et al., 1998). Furthermore, RNA sequencing showed expression of other dopamine receptors in the gill, specifically $D_1$-like

receptors in NECs (Pan et al., 2022). Zebrafish have served as an excellent model to explore the evolutionary origins of dopamine, and $D_2Rs$, and may be particularly useful in clarifying a role for other dopamine receptors in hypoxia signalling.

## Conclusions

The present study has delineated a mechanism by which presynaptic dopamine $D_2Rs$ provide a feedback mechanism that attenuates the chemoreceptor response to hypoxia. We found that activation of presynaptic $D_2Rs$ decreases the NEC $Ca^{2+}$ response to hypoxia via intracellular AC inhibition. We provide evidence for postsynaptic modulation of the $Ca^{2+}$ response to hypoxia via presynaptic $D_2Rs$ and suggest a link between chemoreceptor inhibition by dopamine and modulation of the hypoxic ventilatory response. Our results provide the first evidence of neuromodulation of the hypoxic signal produced by NECs in the gill and suggest that a modulatory role for dopamine in oxygen sensing arose early in vertebrate evolution.

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

## Additional information

### Data availability statement

Raw data that support the findings of this study are available from the corresponding author upon reasonable request.

### Competing interests

None declared.

### Author contributions

M.R and M.G.J conceived and designed the experimental approach, and were involved in interpretation of the results. M.R performed the experiments, data analysis, preparation of the figures and wrote the manuscript. M.R and M.G.J edited and revised the manuscript. Both authors approved the final version of the manuscript submitted for publication and agree to be accountable for all aspects of the work.

### Funding

This work was supported by the Natural Sciences and Engineering Research Council of Canada through Discovery Grants to MGJ (2018-05571 and 2024-03908).

### Keywords

calcium, chemoreceptor, D$_2$, dopamine, hypoxia, NEC, neuroepithelial cell, zebrafish

### Supporting information

Additional supporting information can be found online in the Supporting Information section at the end of the HTML view of the article. Supporting information files available:

**Peer Review History**

