## [Peer Review History · The Journal of Physiology]

Oxygen chemoreceptor inhibition by dopamine D2 receptors in isolated zebrafish gills

Maddison Reed and Michael G Jonz
DOI: 10.1113/JP287824

Corresponding author(s): Michael Jonz (mjonz@uottawa.ca)

The following individual(s) involved in review of this submission have agreed to reveal their identity: Colin A. Nurse (Referee #2)

Review Timeline:

Submission Date:	08-Oct-2024
Editorial Decision:	22-Nov-2024
Revision Received:	24-Jan-2025
Editorial Decision:	05-Feb-2025
Revision Received:	06-Feb-2025
Editorial Decision:	10-Feb-2025
Revision Received:	10-Feb-2025
Accepted:	13-Feb-2025

Senior Editor: Harold Schultz

Reviewing Editor: Andrew Holmes

Transaction Report:

Dear Dr Jonz,

Re: JP-RP-2024-287824 "Oxygen chemoreceptor inhibition by dopamine D2 receptors in isolated zebrafish gills" by Maddison Reed and Michael G Jonz

Thank you for submitting your manuscript to The Journal of Physiology. It has been assessed by a Reviewing Editor and by 2 expert referees and we are pleased to tell you that it is acceptable for publication following satisfactory revision.

REVISION CHECKLIST:

We look forward to receiving your revised submission.

Yours sincerely,

Harold Schultz
Senior Editor
The Journal of Physiology

REQUIRED ITEMS

- Author photo and profile. First or joint first authors are asked to provide a short biography (no more than 100 words for one author or 150 words in total for joint first authors) and a portrait photograph. These should be uploaded and clearly labelled together in a Word document with the revised version of the manuscript. See Information for Authors for further details.
- Your manuscript must include a complete Additional Information section, including competing interests; funding; author contributions and acknowledgements.
- Please upload separate high-quality figure files via the submission form.
- Please ensure that the Article File you upload is a Word file.
- Papers must comply with the Statistics Policy: https://jp.msubmit.net/cgi-bin/main.plex?form_type=display_requirements#statistics.

In summary:

- If $n \leq 30$, all data points must be plotted in the figure in a way that reveals their range and distribution. A bar graph with data points overlaid, a box and whisker plot or a violin plot (preferably with data points included) are acceptable formats.
- If $n > 30$, then the entire raw dataset must be made available either as supporting information, or hosted on a not-for-profit repository, e.g. FigShare, with access details provided in the manuscript.
- 'n' clearly defined (e.g. x cells from y slices in z animals) in the Methods. Authors should be mindful of pseudoreplication.
- All relevant 'n' values must be clearly stated in the main text, figures and tables.
- The most appropriate summary statistic (e.g. mean or median and standard deviation) must be used. Standard Error of the Mean (SEM) alone is not permitted.

- Exact p values must be stated. Authors must not use 'greater than' or 'less than'. Exact p values must be stated to three significant figures even when 'no statistical significance' is claimed.

- Please include an Abstract Figure file, as well as the Figure Legend text within the main article file. The Abstract Figure is a piece of artwork designed to give readers an immediate understanding of the research and should summarise the main conclusions. If possible, the image should be easily 'readable' from left to right or top to bottom. It should show the physiological relevance of the manuscript so readers can assess the importance and content of its findings. Abstract Figures should not merely recapitulate other figures in the manuscript. Please try to keep the diagram as simple as possible and without superfluous information that may distract from the main conclusion(s). Abstract Figures must be provided by authors no later than the revised manuscript stage and should be uploaded as a separate file during online submission labelled as File Type 'Abstract Figure'. Please also ensure that you include the figure legend in the main article file. All Abstract Figures should be created using BioRender. Authors should use The Journal's premium BioRender account to export high-resolution images. Details on how to use and access the premium account are included as part of this email.

Reviewing Editor's comments:

Thank you for taking the time to submit to the Journal of Physiology, we appreciate the amount of effort that goes into such submissions. Overall, both reviewers were extremely positive about the manuscript in terms of the elegance of the experimental approach and the importance of the research findings. I'd like to add that the presentation of the data is of extremely high quality and is incredibly clear. Thank you for this. I also agree that the idea of dopamine being an important evolutionary conserved neuromodulator of O₂ sensing arising early on in vertebrates is highly important and likely to be high interest to the field and also the wider scientific community. That said, the reviewers still raises some important points that need to be addressed. In addition I have a couple of points to need to be considered-

Please could you provide clear ethical approval for these experiments- was there a specific Animal Welfare and Ethical Review Body at the University that approved these experiments and were they approved by a Canadian governmental regulatory body. I can't quite make out if this is the case at the moment. Also, can authors just make it clear that the whole gill baskets were removed after confirmation of death (maybe include how death was confirmed as well).

Please can all data be presented with the SD rather than SEM, and instead of stars and n.s. could authors please provide the actual p values in the figures and text throughout.

Please can you state in the figure legend what the n numbers are referring to exactly - number of cells or number of fish. Also in the figure legends, where abbreviations are used please can you spell out what the abbreviation means upon its first use.

Figure 6A- the experiments here seem to follow a different experimental protocol compared to the others presented in the manuscript. Was there a specific reason for this. The mean data is still presented as the others so I'm a little confused about what was actually done here. Could you pls clarify.

Senior Editor:

Comments to ensure the paper complies with the Statistics Policy:

Data summaries must be defined as mean with standard deviation (standard error is not acceptable), or median with quartiles.

State sample size for each experimental group and whether numbers are replications of the same experimental sample, multiple samples taken from the same source, independent samples taken from a population, or a combination of these replicates. Indicate when samples were excluded for any reason.

The exact p values must be stated in text, tables, and in figures to three significant figures even when 'no statistical significance' is being reported. As an exception, stating $p < 0.001$ is permitted. Do not use symbols to denote statistical significance.

Comments to the authors:

Thank you for submission of your research article to the Journal of Physiology for consideration. The article has been reviewed by experts in the field and found to require revision to address all of the concerns raised. Please address all comments from the external referees and reviewing editor as well as addressing the list of requirements or publication in the journal included in this letter and well a conforming to the Journal's policies for rigour and reproducibility.

Please consult the Journal policy on Rigour and Reproducibility provided with the following link:<https://physoc.onlinelibrary.wiley.com/pb-assets/hub-assets/physoc/documents/TJP-Rigour-and-Reproducibility-Requirements-1724673661727.pdf>

Please ensure the manuscript conforms to these guidelines. Notable deficiencies were found (not an exclusive list).

Methods:

The method of termination of the animals must be be stated.

Describe the controls used in the study. For genetically modified animals, wildtype controls including background and back-crossing must be defined.

Include precise details of all experimental procedures on animals including drug formulations and dose, route and frequency of administration and outcome measures. It was not clear how hypoxia was administered, the level(s) of hypoxia imposed, the time period(s) for baseline, experimental intervention, and recovery between trials if repeated trials were administered.

Statistics:

Data summaries must be defined as mean with standard deviation (standard error is not acceptable), or median with quartiles.

State sample size for each experimental group and whether numbers are replications of the same experimental sample, multiple samples taken from the same source, independent samples taken from a population, or a combination of these replicates. Indicate when samples were excluded for any reason. N for statistical analysis should represent true independent samples and not replicates from the same sample (pseudo replication).

The exact p values must be stated in text, tables, and in figures to three significant figures even when 'no statistical significance' is being reported. As an exception, stating $p < 0.001$ is permitted. Do not use symbols to denote statistical significance.

Referee #1:

This is a beautifully conducted study demonstrating the role of dopamine in the control of the hypoxic ventilatory response in fish. The study is well conceived, designed and executed. Overall, I think that this work timely and useful. I especially like the proof of concept figure (Fig. 2) with consistent repeatability of the hypoxic response. The development of a whole-gill preparation using an endogenous Ca²⁺ indicator to capture closer to in vivo conditions is a powerful new technique that will add to the field of chemosensing in vertebrates.

Referee #2:

In this paper, the authors used a transgenic zebrafish line in which the Ca²⁺ reporter, GCaMP, was targeted to O₂ chemoreceptors (NECs) and endogenous local neurons in the zebrafish gill to investigate the role of dopamine (DA) and DA-D2 receptors (D2R) in sensory transmission. In the isolated, intact whole-gill preparations, they show that in presynaptic NECs hypoxia elicits intracellular Ca²⁺ increases arising from both extracellular and intracellular sources. These hypoxia-induced Ca²⁺ responses were reversibly suppressed by DA agonists acting via D2R, and enhanced by D2R antagonists. Hypoxia also elicited Ca²⁺ elevations in the soma of nearby postsynaptic neurons (ChNs), and several lines of evidence suggested these Ca²⁺ responses were indirect, arising from synaptic transmission from presynaptic NECs. Because the postsynaptic ChNs express the DA transporter and are a potential source of gill DA, they conclude that the DA- D2R pathway provides a negative feedback mechanism that attenuates the NEC chemoreceptor response to hypoxia. Given that a similar role for DA-D2R has been established in the O₂ sensing mammalian carotid body, the data suggest that a modulatory role for dopamine in oxygen sensing arose early in vertebrate evolution.

Critique:

General

Overall this is an elegant and impactful study using a tractable preparation to study O₂ sensing and chemotransmission, as well as the inhibitory role of DA, in the zebrafish gill. A clear strength of the study is the repeatability and robustness of the hypoxia-induced Ca²⁺ signals recorded in the presynaptic receptor cells (NECs) and postsynaptic sensory/relay neurons (ChNs) in a novel ex vivo, intact gill preparation. Moreover, their ability to simultaneously record Ca²⁺ responses during chemostimulation in both cell types is commendable. In many cases, reversibility of the chemosensory response was demonstrated after several stimulus and drug applications, thereby adding confidence in the results. Confocal immunofluorescence studies nicely complemented the main findings. The role of DA acting via D2R as a negative feedback inhibitory pathway in the gill was amply demonstrated, with the use of appropriate agonists and antagonists. However, candidates for the excitatory chemosensory mechanism (e.g. 5-HT) were not addressed, nor were the downstream effectors mediating the inhibitory DA-D2R pathway. Though the potential for direct O₂ sensing by ChNs was addressed, and evidence against such a mechanism was presented, other more compelling approaches could have been considered, as discussed below. Addressing these points would considerably increase the impact of the paper in the opinion of this reviewer. I submit the following comments for the authors' consideration.

Major Comments:

1. In this same GCaMP transgenic zebrafish line, Sebe et al. detected Ca²⁺ elevations in sensory afferent terminals during chemical irritation (J Neurosci 2017 Jun 21;37(25):6162-6175). While convincing post hoc immunofluorescence staining was used to confirm cell identity in the present study, the authors should comment on why their reported Ca²⁺ signals in NECs would exclude 'contamination' from closely abutting nerve endings.

2. The authors addressed the important issue whether ChNs acted as postsynaptic afferent neurons that received synaptic input from presynaptic O₂ chemoreceptors (NECs) and, consequently, were not themselves directly sensitive to hypoxia. To support this model, they carried out experiments aimed to remove functional connections (in an irreversible manner) between NECs and ChNs, before testing the effects of hypoxia. These experiments included 'filament transection' to mechanically isolate proximal ChNs from NECs and administration of 6-OHDA to chemically destroy dopaminergic nerve terminals contacting NECs. While these experiments supported the model, I believe other less-invasive approaches could have been considered, allowing increased sampling and easier data interpretation. For example, the use of TTX, a well-characterized and readily reversible blocker of voltage-gated Na⁺ channels (and therefore afferent action potential propagation in ChNs) would be of interest. The rationale here is that NECs appear to lack voltage-gated Na⁺ channels and it is known that zebrafish sensory axons express TTX-sensitive Na⁺ channels (Won et al. 2012 PLoS One. 2012 Aug 3;7(8):e42602). Thus, in the presence of TTX, hypoxia-induced Ca²⁺ signals should be detectable in NECs but not ChNs according to the model and, importantly, responses in both cell types should recover after drug wash-out.

A second approach might also provide additional information that would increase the impact of the study. In the discussion, the authors note the excitatory neurotransmitter(s) at NEC-ChN synapse is(are) not firmly established, however, the leading candidate 5-HT is synthesized by NECs. In addition, a previous transcriptome study from this lab (Pan et al. 2022) indicated high expression of 5-HT_{3A}-like receptors in gill neurons, which likely include the ChN population. This begs the question: Is the hypoxia-induced Ca²⁺ signal in ChNs inhibited by 5-HT_{3A} antagonists, e.g. MDL72222? A positive answer would be a welcomed addition, not only providing new information on the possible identity of the excitatory neurotransmitter(s), but also adding an elegant way of showing ChNs are postsynaptic and not directly sensitive to hypoxia.

A third approach was considered by the authors though it could have been expanded further. For example, on page 11, the authors report that in dual recordings of Ca²⁺ signals during hypoxia, "NECs responded first with an increase in [Ca²⁺]_i, followed by a [Ca²⁺]_i response in ChNs after a latency of 20-30 s (Fig. 9D)". This is supportive evidence that the ChN soma response is postsynaptic and mediated via synaptic transmission from NECs to ChN nerve endings. The authors should quantify these data providing 'n' values and mean (+/- sem) latency times. A nice addition would be to show whether this latency is temperature dependent, i.e. increasing markedly at reduced temperatures, given the well-known temperature sensitivity (Q₁₀) of synaptic transmission and action potential propagation. Similar to the TTX and 5-HT_{3AR} blocker-experiments suggested above, this approach also has the advantage that the ChNs are not exposed to potentially harmful results (mechanical or chemical) and response reversibility should be readily demonstrated.

3. The level of hypoxia used was 25 mmHg throughout. Is there a threshold level of hypoxia, below which neurotransmission is not detected in ChNs but Ca²⁺ elevation can still be detected in NECs?

4. On p13 of Discussion, the authors may note that D₂R may signal via pathways other than those involving AC and cAMP. Given that it's reasonable to focus on AC-cAMP in the present study, it would be helpful for the reader if the authors could show a proposed model of the sensory synapse during hypoxia, involving pre- and post-synaptic elements, as well as the main players relevant to the study, e.g. DA, D₂R, AC, cAMP, DAT. What about possible downstream effectors of cAMP? Can background K⁺ channels in NECs be modulated via this pathway, leading to changes in intracellular Ca²⁺? Related to this, does the D₂R pathway affect the hypoxia response via intra or extracellular Ca²⁺ pathway? In Fig. 6, is SQ22536 affecting stores Ca²⁺ or extracellular Ca²⁺ pathway or both?

Minor Comments:

-Under Key points item 1; note that two-photon Ca²⁺ imaging of oxygen chemoreceptors in intact or mouse carotid body (CB) using GCaMP has previously been reported (see Timón-Gómez et al. eLife. 2022 Oct 18;11:e78915).

-Abstract line 2: insert 'mammalian' before carotid body; also it's questionable whether DA can be described as the 'first' neurotransmitter in the CB. It's more accurate to say..... DA is best described and most abundant. Historically, and though controversial, ACh received the most attention as an excitatory CB neurotransmitter in the 1950's.

-When referring to the D₂ receptor throughout the manuscript, the abbreviation D₂R is more informative than D₂.

-Introduction line 11; replace 'mediating' by 'modulating'.

-In Fig. 1, aren't ChN nerve endings also labeled with SV2 in addition to NECs? There appears to be much more magenta SV2 staining than GFP staining. Can the authors comment on this? Is there variable GFP staining in NECs? In Fig. 1E, what

is the green 'cell/structure' not labelled with 5-HT? Could these be nerve endings which should stain for SV2 but not 5-HT? Arrows will be helpful in these photomicrographs.

-Fig. 3B. Is hypoxia-3 in (B) data from the 3rd or 4th trace in sequence shown in A; or is right bin in (B) hypoxia-4?

-Fig. 4 legend last line. The authors state " Blocking both intracellular and extracellular Ca²⁺ resulted in the largest change in the Ca²⁺ response to hypoxia (F/F₀)". This is not obvious from the graph, especially when compared with nifedipine. In Fig. 4B, what is origin of residual Ca²⁺ response to hypoxia in presence of dantrolene + 0 Ca²⁺?

-Fig. 5, do the authors know whether blockade of DAT, thereby increasing extracellular DA levels, reduces the NEC response to hypoxia? This could be a nice complement to the dramatic effect of quinpirole in suppressing the hypoxic response.

-In nerve cut and 6-OHDA experiments (Fig. 8D-F), provide 'n' values and mean (+/-) F/F₀ data for high K⁺/hypoxia stimuli. Also, how long after these treatments were Ca²⁺ signals obtained from ChNs exposed to hypoxia?

- Fig. 9 D legend. Purple and blue colors are too similar. Better contrast will be helpful.

- In Fig. 9, the authors aimed to link D2R activity in NECs with the Ca²⁺ response in postsynaptic ChNs. To show functional absence of D2R on ChNs, they report that quinpirole failed to reduce Ca²⁺ response to high K⁺ stimulus. This is not a compelling argument since any rapidly inactivating Ca²⁺ channels (T- or N-type, if present?) will be closed in high K⁺ and not subject to modulation.

-Discussion: Given the demonstrated role of both extra- and intra-cellular Ca²⁺ in NEC O₂ sensing, the authors might want to link their results to studies on O₂ sensing mechanisms in carotid body type I cells and the potential role of Ca²⁺ oscillations mediated by ER/SOCE mechanisms and sustained by Ca²⁺ entry through voltage-gated Ca²⁺ channels (Kim et al 2020. AJP Cell Physiol 318: C430-C438).

END OF COMMENTS

In this paper, the authors used a transgenic zebrafish line in which the Ca²⁺ reporter, GCaMP, was targeted to O₂ chemoreceptors (NECs) and endogenous local neurons in the zebrafish gill to investigate the role of dopamine (DA) and DA-D₂ receptors (D₂R) in sensory transmission. In the isolated, intact whole-gill preparations, they show that in presynaptic NECs hypoxia elicits intracellular Ca²⁺ increases arising from both extracellular and intracellular sources. These hypoxia-induced Ca²⁺ responses were reversibly suppressed by DA agonists acting via D₂R, and enhanced by D₂R antagonists. Hypoxia also elicited Ca²⁺ elevations in the soma of nearby postsynaptic neurons (ChNs), and several lines of evidence suggested these Ca²⁺ responses were indirect, arising from synaptic transmission from presynaptic NECs. Because the postsynaptic ChNs express the DA transporter and are a potential source of gill DA, they conclude that the DA-D₂R pathway provides a negative feedback mechanism that attenuates the NEC chemoreceptor response to hypoxia. Given that a similar role for DA-D₂R has been established in the O₂ sensing mammalian carotid body, the data suggest that a modulatory role for dopamine in oxygen sensing arose early in vertebrate evolution.

Critique:

General

Overall this is an elegant and impactful study using a tractable preparation to study O₂ sensing and chemotransmission, as well as the inhibitory role of DA, in the zebrafish gill. A clear strength of the study is the repeatability and robustness of the hypoxia-induced Ca²⁺ signals recorded in the presynaptic receptor cells (NECs) and postsynaptic sensory/relay neurons (ChNs) in a novel *ex vivo*, intact gill preparation. Moreover, their ability to simultaneously record Ca²⁺ responses during chemostimulation in both cell types is commendable. In many cases, reversibility of the chemosensory response was demonstrated after several stimulus and drug applications, thereby adding confidence in the results. Confocal immunofluorescence studies nicely complemented the main findings. The role of DA acting via D₂R as a negative feedback inhibitory pathway in the gill was amply demonstrated, with the use of appropriate agonists and antagonists. However, candidates for the excitatory chemosensory mechanism (e.g. 5-HT) were not addressed, nor were the downstream effectors mediating the inhibitory DA-D₂R pathway. Though the potential for direct O₂ sensing by ChNs was addressed, and evidence against such a mechanism was presented, other more compelling approaches could have been considered, as discussed below. Addressing these points would considerably increase the impact of the paper in the opinion of this reviewer. I submit the following comments for the authors' consideration.

Major Comments:

1. In this same GCaMP transgenic zebrafish line, *Sebe et al.* detected Ca²⁺ elevations in sensory afferent terminals during chemical irritation (J Neurosci 2017 Jun 21;37(25):6162-6175). While convincing *post hoc* immunofluorescence staining was used to confirm cell identity In the present study, the authors should comment on why their reported Ca²⁺ signals in NECs would exclude 'contamination' from closely abutting nerve endings.

2. The authors addressed the important issue whether ChNs acted as postsynaptic afferent neurons that received synaptic input from presynaptic O₂ chemoreceptors (NECs) and, consequently, were *not* themselves directly sensitive to hypoxia. To support this model, they carried out experiments aimed to remove functional connections (in an *irreversible* manner) between NECs and ChNs, before testing the effects of hypoxia. These experiments included 'filament transection' to mechanically isolate proximal ChNs from NECs and administration of 6-OHDA to chemically destroy dopaminergic nerve terminals contacting NECs. While these experiments supported the model, I believe other less-invasive approaches could have been considered, allowing increased sampling and easier data interpretation. For example, the use of TTX, a well-characterized and readily reversible blocker of voltage-gated Na⁺ channels (and therefore afferent action potential propagation in ChNs) would be of interest. The rationale here is that NECs appear to lack voltage-gated Na⁺ channels and it is known that zebrafish sensory axons express TTX-sensitive Na⁺ channels (Won et al. 2012 PLoS One. 2012 Aug 3;7(8):e42602). Thus, in the presence of TTX, hypoxia-induced Ca²⁺ signals should be detectable in NECs but not ChNs according to the model and, importantly, responses in both cell types should recover after drug wash-out.

A second approach might also provide additional information that would increase the impact of the study. In the discussion, the authors note the excitatory neurotransmitter(s) at NEC-ChN synapse is(are) not firmly established, however, the leading candidate 5-HT is synthesized by NECs. In addition, a previous transcriptome study from this lab (Pan et al. 2022) indicated high expression of 5-HT_{3A}-like receptors in gill neurons, which likely include the ChN population. This begs the question: Is the hypoxia-induced Ca²⁺ signal in ChNs inhibited by 5-HT_{3A} antagonists, e.g. MDL72222? A positive answer would be a welcomed addition, not only providing new information on the possible identity of the excitatory neurotransmitter(s), but also adding an elegant way of showing ChNs are postsynaptic and not directly sensitive to hypoxia.

A third approach was considered by the authors though it could have been expanded further. For example, on page 11, the authors report that in dual recordings of Ca²⁺ signals during hypoxia, "*NECs responded first with an increase in [Ca²⁺]_i, followed by a [Ca²⁺]_i response in ChNs after a latency of 20-30 s (Fig. 9D)*". This is supportive evidence that the ChN soma response is postsynaptic and mediated via synaptic transmission from NECs to ChN nerve endings. The authors should quantify these data providing 'n' values and mean (+/- sem) latency times. A nice addition would be to show whether this latency is temperature dependent, i.e. increasing markedly at reduced temperatures, given the well-known temperature sensitivity (Q₁₀) of synaptic transmission and action potential propagation. Similar to the TTX and 5-HT_{3AR} blocker-experiments suggested above, this approach also has the advantage that the ChNs are not exposed to potentially harmful insults (mechanical or chemical) and response reversibility should be readily demonstrated.

3. The level of hypoxia used was 25 mmHg throughout. Is there a threshold level of hypoxia, below which neurotransmission is not detected in ChNs but Ca²⁺ elevation can still be detected in NECs?

4. On p13 of Discussion, the authors may note that D2R may signal via pathways other than those involving AC and cAMP. Given that it's reasonable to focus on AC-cAMP in the present study, it would be helpful for the reader if the authors could show a proposed model of the sensory synapse during hypoxia, involving pre- and post-synaptic elements, as well as the main players relevant to the study, e.g. DA, D2R, AC, cAMP, DAT. What about possible downstream effectors of cAMP? Can background K⁺ channels in NECs be modulated via this pathway, leading to changes in intracellular Ca²⁺? Related to this, does the D2R pathway affect the hypoxia response via intra or extracellular Ca²⁺ pathway? In Fig. 6, is SQ22536 affecting stores Ca²⁺ or extracellular Ca²⁺ pathway or both?

Minor Comments:

-Under Key points item 1; note that two-photon Ca²⁺ imaging of oxygen chemoreceptors in intact or mouse carotid body (CB) using GCaMP has previously been reported (see Timón-Gómez et al. eLife. 2022 Oct 18;11:e78915).

-Abstract line 2: insert 'mammalian' before carotid body; also it's questionable whether DA can be described as the 'first' neurotransmitter in the CB. It's more accurate to say..... DA is best described and most abundant. Historically, and though controversial, ACh received the most attention as an excitatory CB neurotransmitter in the 1950's.

-When referring to the D2 receptor throughout the manuscript, the abbreviation D2R is more informative than D2.

-Introduction line 11; replace 'mediating' by 'modulating'.

-In Fig. 1, aren't ChN nerve endings also labeled with SV2 in addition to NECs? There appears to be much more magenta SV2 staining than GFP staining. Can the authors comment on this? Is there variable GFP staining in NECs? In Fig. 1E, what is the green 'cell/structure' not labelled with 5-HT? Could these be nerve endings which should stain for SV2 but not 5-HT? Arrows will be helpful in these photomicrographs.

-Fig. 3B. Is hypoxia-3 in (B) data from the 3rd or 4th trace in sequence shown in A; or is right bin in (B) hypoxia-4?

-Fig. 4 legend last line. The authors state "*Blocking both intracellular and extracellular Ca²⁺ resulted in the largest change in the Ca²⁺ response to hypoxia ($\Delta F/F_0$)*". This is not obvious from the graph, especially when compared with nifedipine. In Fig. 4B, what is origin of residual Ca²⁺ response to hypoxia in presence of dantrolene + 0 Ca²⁺?

-Fig. 5, do the authors know whether blockade of DAT, thereby increasing extracellular DA levels, reduces the NEC response to hypoxia? This could be a nice complement to the dramatic effect of quinpirole in suppressing the hypoxic response.

-In nerve cut and 6-OHDA experiments (Fig. 8D-F), provide 'n' values and mean (+/-) F/Fo data for high K+/hypoxia stimuli. Also, how long after these treatments were Ca²⁺ signals obtained from ChNs exposed to hypoxia?

- Fig. 9 D legend. Purple and blue colors are too similar. Better contrast will be helpful.

- In Fig. 9, the authors aimed to link D2R activity in NECs with the Ca²⁺ response in postsynaptic ChNs. To show functional absence of D2R on ChNs, they report that quinpirole failed to reduce Ca²⁺ response to high K⁺ stimulus. This is not a compelling argument since any rapidly inactivating Ca²⁺ channels (T- or N-type, if present?) will be closed in high K⁺ and not subject to modulation.

-Discussion: Given the demonstrated role of both extra- and intra-cellular Ca²⁺ in NEC O₂ sensing, the authors might want to link their results to studies on O₂ sensing mechanisms in carotid body type I cells and the potential role of Ca²⁺ oscillations mediated by ER/SOCE mechanisms and sustained by Ca²⁺ entry through voltage-gated Ca²⁺ channels (Kim et al 2020. AJP Cell Physiol 318: C430-C438).

>> Author response. Note: line numbers indicated here are reflected in the “track changes” version of the manuscript.

REVIEWING EDITOR COMMENTS

Thank you for taking the time to submit to the Journal of Physiology, we appreciate the amount of effort that goes into such submissions. Overall, both reviewers were extremely positive about the manuscript in terms of the elegance of the experimental approach and the importance of the research findings. I'd like to add that the presentation of the data is of extremely high quality and is incredibly clear. Thank you for this. I also agree that the idea of dopamine being an important evolutionary conserved neuromodulator of O₂ sensing arising early on in vertebrates is highly important and likely to be high interest to the field and also the wider scientific community. That said, the reviewers still raises some important points that need to be addressed. In addition I have a couple of points to need to be considered-

>> We thank the reviewing editor for their helpful comments.

Please could you provide clear ethical approval for these experiments- was there a specific Animal Welfare and Ethical Review Body at the University that approved these experiments and were they approved by a Canadian governmental regulatory body. I can't quite make out if this is the case at the moment. Also, can authors just make it clear that the whole gill baskets were removed after confirmation of death (maybe include how death was confirmed as well).

>> Some of this information is stated on lines 133-137 "All procedures for animal use and euthanasia were carried out in accordance with institutional guidelines according to protocol BL-3666, and guidelines provided by the Canadian Council on Animal Care." In addition, we have updated our methods to clarify that "Adult zebrafish were euthanized by concussion and rapid decapitation. Euthanasia was confirmed by cessation of movement and breathing" on line 132, and "Whole gill baskets were removed from euthanized animals" on line 151.

Please can all data be presented with the SD rather than SEM, and instead of stars and n.s. could authors please provide the actual p values in the figures and text throughout.

>> We have changed all data summaries to be presented as means with standard deviation and have updated the methods and results sections of the manuscript to reflect these changes.

Please can you state in the figure legend what the n numbers are referring to exactly - number of cells or number of fish. Also in the figure legends, where abbreviations are used please can you spell out what the abbreviation means upon its first use.

>> We have added “cells” in the figure legends/throughout the text and have added some extra information on sampling in the methods line 232.

Figure 6A- the experiments here seem to follow a different experimental protocol compared to the others presented in the manuscript. Was there a specific reason for this. The mean data is still presented as the others so I'm a little confused about what was actually done here. Could you pls clarify.

>> In Fig. 6A we slightly modified our experimental protocol to observe the effects of SQ22536 within a single bout of hypoxia. The summary data shows mean (\pm SD) F/F_0 during the first two minutes of hypoxia exposure (Hypoxia-1), the reduction in mean (\pm SD) F/F_0 during the entire duration of SQ22536 exposure (Hypoxia-2 + SQ22536) and mean (\pm SD) F/F_0 during the last two minutes of hypoxia exposure (Hypoxia-3). We have updated the figure caption to clarify this protocol.

COMMENTS FROM SENIOR EDITOR:

Data summaries must be defined as mean with standard deviation (standard error is not acceptable), or median with quartiles.

>> We have changed all data summaries to be presented as means with standard deviation and have updated the methods and results sections of the manuscript to reflect these changes.

State sample size for each experimental group and whether numbers are replications of the same experimental sample, multiple samples taken from the same source, independent samples taken from a population, or a combination of these replicates. Indicate when samples were excluded for any reason.

>> To clarify sampling, we have included additional information in the methods line 216: "For all reported data, sample size (n) refers to individual cells. While multiple gill arches were assessed per animal, only one cell from each arch was included in the analysis to avoid repeated exposures or treatments in the same tissue. Throughout this study, hypoxic responses from a total of 73 cells were recorded from 41 adult zebrafish."

The exact p values must be stated in text, tables, and in figures to three significant figures even when 'no statistical significance' is being reported. As an exception, stating $p < 0.001$ is permitted. Do not use symbols to denote statistical significance.

>> We have added exact p values to three significant figures in all figures and have updated the manuscript to reflect these changes.

REVIEWER COMMENTS

Reviewer: 1

General comments for authors:

This is a beautifully conducted study demonstrating the role of dopamine in the control of the hypoxic ventilatory response in fish. The study is well conceived, designed and executed. Overall, I think that this work timely and useful. I especially like the proof of concept figure (Fig. 2) with consistent repeatability of the hypoxic response. The development of a whole-gill preparation using an endogenous Ca²⁺ indicator to capture closer to in vivo conditions is a powerful new technique that will add to the field of chemosensing in vertebrates.

>> We thank the reviewer for their comments on our manuscript.

Reviewer: 2

General comments for authors:

Overall this is an elegant and impactful study using a tractable preparation to study O₂ sensing and chemotransmission, as well as the inhibitory role of DA, in the zebrafish gill. A clear strength of the study is the repeatability and robustness of the hypoxia-induced Ca²⁺ signals recorded in the presynaptic receptor cells (NECs) and postsynaptic sensory/relay neurons (ChNs) in a novel ex vivo, intact gill preparation. Moreover, their ability to simultaneously record Ca²⁺ responses during chemostimulation in both cell types is commendable. In many cases, reversibility of the chemosensory response was demonstrated after several stimulus and drug applications, thereby adding confidence in the results. Confocal immunofluorescence studies nicely complemented the main findings. The role of DA acting via D2R as a negative feedback inhibitory pathway in the gill was amply demonstrated, with the use of appropriate agonists and antagonists. However, candidates for the excitatory chemosensory mechanism (e.g. 5-HT) were not addressed, nor were the downstream effectors mediating the inhibitory DA-D2R pathway. Though the potential for direct O₂ sensing by ChNs was addressed, and evidence against such a mechanism was presented, other more compelling approaches could have been considered, as discussed below. Addressing these points would considerably increase the impact of the paper in the opinion of this reviewer. I submit the following comments for the authors' consideration.

>> We thank the reviewer for taking the time to provide valuable feedback on our manuscript.

Major comments:

In this same GCaMP transgenic zebrafish line, Sebe et al. detected Ca²⁺ elevations in sensory afferent terminals during chemical irritation (J Neurosci 2017 Jun 21;37(25):6162-6175). While convincing post hoc immunofluorescence staining was used to confirm cell identity In the present study, the authors should comment on why their reported Ca²⁺ signals in NECs would exclude 'contamination' from closely abutting nerve endings.

>> There are a few key differences between the preparations by Sebe et al. and ours in this study that exclude observations of nerve terminal activity. Sebe et al. report Ca²⁺ elevations from Tg(elavl3:GCaMP5G) zebrafish using light sheet microscopy. Our preparation uses a different calcium probe – GCaMP6s, and lower resolution imaging system. Since GCaMP6s is a cytosolic probe and mainly localized to the soma, we have not been able to observe nerve terminals in any of our preparations.

2. The authors addressed the important issue whether ChNs acted as postsynaptic afferent neurons that received synaptic input from presynaptic O₂ chemoreceptors (NECs) and, consequently, were not themselves directly sensitive to hypoxia. To support this model, they carried out experiments aimed to remove functional connections (in an irreversible manner) between NECs and ChNs, before testing the effects of hypoxia. These experiments included 'filament transection' to mechanically isolate proximal ChNs from NECs and administration of 6-OHDA to chemically destroy dopaminergic nerve terminals contacting NECs. While these experiments supported the model, I believe other less-invasive approaches could have been considered, allowing increased sampling and easier data interpretation. For example, the use of TTX, a well-characterized and readily reversible blocker of voltage-gated Na⁺ channels (and therefore afferent action potential propagation in ChNs) would be of interest. The rationale here is that NECs appear to lack voltage-gated Na⁺ channels and it is known that zebrafish sensory axons express TTX-sensitive Na⁺ channels (Won et al. 2012 PLoS One. 2012 Aug 3;7(8):e42602). Thus, in the presence of TTX, hypoxia-induced Ca²⁺ signals should be detectable in NECs but not ChNs according to the model and, importantly, responses in both cell types should recover after drug wash-out.

>>This is an interesting idea that would support our argument that the hypoxic signal is transferred from NEC to ChN, but it would not directly address whether ChNs can respond to hypoxia in the absence of a functional chemoreceptor, i.e. have intrinsic oxygen sensitivity. The rationale for our ablation and 6-OHDA experiments was to simultaneously show that stimulation of the NEC leads to a response in the postsynaptic ChN, but also that

ChNs cannot themselves mount a response to hypoxia without the NEC. We believe our approach provides a more convincing result but will consider the TTX experiments for an upcoming study.

A second approach might also provide additional information that would increase the impact of the study. In the discussion, the authors note the excitatory neurotransmitter(s) at NEC-ChN synapse is(are) not firmly established, however, the leading candidate 5-HT is synthesized by NECs. In addition, a previous transcriptome study from this lab (Pan et al. 2022) indicated high expression of 5-HT3A-like receptors in gill neurons, which likely include the ChN population. This begs the question: Is the hypoxia-induced Ca²⁺ signal in ChNs inhibited by 5-HT3A antagonists, e.g. MDL72222? A positive answer would be a welcomed addition, not only providing new information on the possible identity of the excitatory neurotransmitter(s), but also adding an elegant way of showing ChNs are postsynaptic and not directly sensitive to hypoxia.

>>We thank the reviewer for this suggestion. They are correct that 5-HT3A receptors are expressed in gill neurons. In confidence, we can say that our preliminary evidence from a parallel study indicates that functional 5-HT receptors are not present at the NEC-ChN synapse, but appear to form part of a separate pathway in the gill that we are currently characterizing. For now, we have included a statement on line 416 indicating: "The excitatory neurotransmitter at the NEC-ChN synapse has yet to be elucidated. Candidates include acetylcholine, which is excitatory in the carotid body and is found in the zebrafish gill (Zachar et al., 2017b) and 5-HT, which is present in most NECs in zebrafish (Jonz & Nurse 2003)."

A third approach was considered by the authors though it could have been expanded further. For example, on page 11, the authors report that in dual recordings of Ca²⁺ signals during hypoxia, "NECs responded first with an increase in [Ca²⁺]_i, followed by a [Ca²⁺]_i response in ChNs after a latency of 20-30 s (Fig. 9D)". This is supportive evidence that the ChN soma response is postsynaptic and mediated via synaptic transmission from NECs to ChN nerve endings. The authors should quantify these data providing 'n' values and mean (+/- sem) latency times. A nice addition would be to show whether this latency is temperature dependent, i.e. increasing markedly at reduced temperatures, given the well-known temperature sensitivity (Q₁₀) of synaptic transmission and action potential propagation. Similar to the TTX and 5-HT3AR blocker- experiments suggested above, this approach also has the advantage that the ChNs are not exposed to potentially harmful results (mechanical or chemical) and response reversibility should be readily demonstrated.

>>This is another interesting idea, but perhaps for future investigations. Fortunately, the order of response was always unidirectional, that is to say that NECs always responded first followed by the ChN response. This not only suggests that these two cell types form a functional synapse, but also that excitatory activation moves only from NEC to ChN. We

have quantified latency times and provided means +/- SD with sample size. This information has been added to the results section (line 325).

3. The level of hypoxia used was 25 mmHg throughout. Is there a threshold level of hypoxia, below which neurotransmission is not detected in ChNs but Ca²⁺ elevation can still be detected in NECs?

>> The reviewer raises a very interesting question. Since it is possible that multiple NECs contact a single chain neuron, it may not be possible to observe such a threshold unless all other NECs can be ablated. Isolating a single unit (one NEC and one ChN) would be a very interesting preparation to try in a future study.

4. On p13 of Discussion, the authors may note that D2R may signal via pathways other than those involving AC and cAMP. Given that it's reasonable to focus on AC-cAMP in the present study, it would be helpful for the reader if the authors could show a proposed model of the sensory synapse during hypoxia, involving pre- and post-synaptic elements, as well as the main players relevant to the study, e.g. DA, D2R, AC, cAMP, DAT. What about possible downstream effectors of cAMP? Can background K⁺ channels in NECs be modulated via this pathway, leading to changes in intracellular Ca²⁺? Related to this, does the D2R pathway affect the hypoxia response via intra or extracellular Ca²⁺ pathway? In Fig. 6, is SQ22536 affecting stores Ca²⁺ or extracellular Ca²⁺ pathway or both?

>> We have added ideas on the downstream effects of cAMP on K⁺ channels, and how they affect intracellular Ca²⁺, in the discussion (line 396). In light of our next prepared manuscript, we feel waiting to produce a proposed model of the sensory synapse will give a bigger and more impactful overview.

Minor Comments:

-Under Key points item 1; note that two-photon Ca²⁺ imaging of oxygen chemoreceptors in intact or mouse carotid body (CB) using GCaMP has previously been reported (see Timón-Gómez et al. eLife. 2022 Oct 18;11:e78915).

>> Key point has been modified.

-Abstract line 2: insert 'mammalian' before carotid body; also it's questionable whether DA can be described as the 'first' neurotransmitter in the CB. It's more accurate to say..... DA is best described and most abundant. Historically, and though controversial, ACh received the most attention as an excitatory CB neurotransmitter in the 1950's.

>> We have made these changes on line 31 of the abstract.

-When referring to the D2 receptor throughout the manuscript, the abbreviation D2R is more informative than D2.

>> We have replaced D2 with D₂R throughout the manuscript.

-Introduction line 11; replace 'mediating' by 'modulating'.

>> We have made this change on line 80.

-In Fig. 1, aren't ChN nerve endings also labeled with SV2 in addition to NECs? There appears to be much more magenta SV2 staining than GFP staining. Can the authors comment on this? Is there variable GFP staining in NECs? In Fig. 1E, what is the green 'cell/structure' not labelled with 5-HT? Could these be nerve endings which should stain for SV2 but not 5-HT? Arrows will be helpful in these photomicrographs.

>> We agree with the reviewer that there appears to be much more magenta (SV2) staining than GFP. We have observed a large variability in the number of cells containing the GCaMP complex throughout development. Specifically, we find a much lower number of GCaMP-positive cells in adulthood, perhaps due to numerous cell division events as the filament tip grows. Since GFP only becomes exposed when the GCaMP complex is activated, we do observe variability in endogenous background GFP across NECs at a given time. In Fig 1E, the green cells not labelled with 5HT may be non-serotonergic SV2-positive cells (Jonz and Nurse 2003) or VAcHT-positive cells (Zachar et al. 2017 Cell Tissue Res) both of which are present in this area of the filament. Arrows have been added to highlight specific overlap between GCaMP and 5HT.

-Fig. 3B. Is hypoxia-3 in (B) data from the 3rd or 4th trace in sequence shown in A; or is right bin in (B) hypoxia-4?

>> Hypoxia-3 in Fig. 3B is data from the 4th trace in the sequence shown. We have changed the x-axis label and figure caption to reflect this clarification.

-Fig. 4 legend last line. The authors state " Blocking both intracellular and extracellular Ca²⁺ resulted in the largest change in the Ca²⁺ response to hypoxia ($\Delta F/F_0$)". This is not obvious from the graph, especially when compared with nifedipine. In Fig. 4B, what is origin of residual Ca²⁺ response to hypoxia in presence of dantrolene + 0 Ca²⁺?

>> We agree that this point could use clarifying in the figure legend. The main point we take away from this result is that the combination of dantrolene + 0 Ca²⁺ together

produced an additive effect compared to dantrolene alone (rather than which column produced the largest change). We have updated Fig. 4 caption to clarify this. In Fig. 4B the residual Ca²⁺ may indicate stores were not completely blocked, as only one concentration of dantrolene was tested.

-Fig. 5, do the authors know whether blockade of DAT, thereby increasing extracellular DA levels, reduces the NEC response to hypoxia? This could be a nice complement to the dramatic effect of quinpirole in suppressing the hypoxic response.

>> Though we did not directly test the blockade of DAT, we believe a similar result to quinpirole would be observed where the NEC response to hypoxia would be reduced.

-In nerve cut and 6-OHDA experiments (Fig. 8D-F), provide 'n' values and mean (+/-) F/F₀ data for high K⁺/hypoxia stimuli. Also, how long after these treatments were Ca²⁺ signals obtained from ChNs exposed to hypoxia?

>> Animals were terminated at the end of their 6-OHDA exposure without recovery. Gill baskets were immediately removed and kept in cold ECS to be evaluated within a few hours of termination. Sample size for these two conditions has been added to Fig. 8 caption and mean ± SD data for hypoxia/high k⁺ stimuli has been added to the results on line 306.

- Fig. 9 D legend. Purple and blue colors are too similar. Better contrast will be helpful.

>> We have changed the purple colour in Fig. 9 to grey for better contrast with blue.

- In Fig. 9, the authors aimed to link D2R activity in NECs with the Ca²⁺ response in postsynaptic ChNs. To show functional absence of D2R on ChNs, they report that quinpirole failed to reduce Ca²⁺ response to high K⁺ stimulus. This is not a compelling argument since any rapidly inactivating Ca²⁺ channels (T- or N-type, if present?) will be closed in high K⁺ and not subject to modulation.

>> We do not feel rapidly inactivating Ca²⁺ channels that will be closed in high K⁺ create an issue with these control experiments. GCaMP6s is a 'slow' type Ca²⁺ reporter that shows changes over several seconds (see Chen et al. 2013. Nature 499:295-300, Fig. 1b) and would not resolve such rapid events produced by T- or N-type channels. In this figure, and throughout the paper, we show Ca²⁺ responses resulting from L-type Ca²⁺ channels.

-Discussion: Given the demonstrated role of both extra- and intra-cellular Ca²⁺ in NEC O₂ sensing, the authors might want to link their results to studies on O₂ sensing mechanisms in carotid body type I cells and the potential role of Ca²⁺ oscillations mediated by ER/SOCE

mechanisms and sustained by Ca²⁺ entry through voltage-gated Ca²⁺ channels (Kim et al 2020. AJP Cell Physiol 318: C430-C438).

>> We thank the reviewing for bringing this paper to our attention. We have added a comparison to the CB on lines 347-350.

Dear Dr Jonz,

Re: JP-RP-2025-287824R1 "Oxygen chemoreceptor inhibition by dopamine D2 receptors in isolated zebrafish gills" by Maddison Reed and Michael G Jonz

Thank you for submitting your manuscript to The Journal of Physiology. It has been assessed by a Reviewing Editor and by 2 expert referees and we are pleased to tell you that it is acceptable for publication following satisfactory revision.

REVISION CHECKLIST:

We look forward to receiving your revised submission.

Yours sincerely,

Harold Schultz
Senior Editor
The Journal of Physiology

REQUIRED ITEMS

- Your manuscript must include a complete Additional Information section, including competing interests; funding; author contributions and acknowledgements.
- The Journal of Physiology funds authors of provisionally accepted papers to use the premium BioRender site to create high resolution schematic figures. Follow this link and enter your details and the manuscript number to create and download figures. Upload these as the figure files for your revised submission. If you choose not to take up this offer, we require figures to be of similar quality and resolution. If you are opting out of this service to authors, state this in the Comments section on the Detailed Information page of the submission form. The link provided should only be used for the purposes of this submission. Authors will be charged for figures created on this premium BioRender account if they are not related to this manuscript submission.
- Papers must comply with the Statistics Policy: https://jp.msubmit.net/cgi-bin/main.plex?form_type=display_requirements#statistics.

In summary:

- If $n \leq 30$, all data points must be plotted in the figure in a way that reveals their range and distribution. A bar graph with data points overlaid, a box and whisker plot or a violin plot (preferably with data points included) are acceptable formats.
- If $n > 30$, then the entire raw dataset must be made available either as supporting information, or hosted on a not-for-profit repository, e.g. FigShare, with access details provided in the manuscript.
- 'n' clearly defined (e.g. x cells from y slices in z animals) in the Methods. Authors should be mindful of pseudoreplication.
- All relevant 'n' values must be clearly stated in the main text, figures and tables.
- The most appropriate summary statistic (e.g. mean or median and standard deviation) must be used. Standard Error of the Mean (SEM) alone is not permitted.
- Exact p values must be stated. Authors must not use 'greater than' or 'less than'. Exact p values must be stated to three significant figures even when 'no statistical significance' is claimed.

Reviewing Editor:

Comments to ensure the paper complies with the Statistics Policy:

Please can the authors make sure to state all the p values in the figures rather than using symbols and n.s. This is in line with the journal policy on stats reporting.

Comments to the Author:

Thank you for taking the time to make the suggested amendments to the manuscript. There is still a small change that is required as outlined by one of the reviewers. In addition, please can the authors make sure to state all the p values in the figures rather than using symbols and n.s. This is in line with the journal policy on stats reporting.

Senior Editor:

Comments to ensure the paper complies with the Statistics Policy:

For ease of reading and comprehension, the Journal strongly suggests displaying the actual P value in the figures.

P values in the figures are in the legend, which is acceptable, but the use of symbols (*) in the figures are not consistently used.

Figure 6A should use (*) to represent $p < 0.05$. (P= 0.01)

Figure 9E should use (*) to represent $p < 0.05$. (P= 0.013)

All other symbols in figures using (**) should represent $P < 0.01$

(Note that symbols in Figures 3-5 are used correctly)

The order of describing stats in the legend of Figure 9E should be reversed with NEC first as in the figure.

Comments to the authors:

Thank you for submission of your revised research article to the Journal of Physiology for consideration. Referee 2 has a remaining concern that must be addressed. In addition, reporting of statistical P values in the figures, and the data availability statement require attention (see below). Please address the remaining comments from the external referee and as well as the concerns below:

1. P values:

For ease of reading and comprehension, the Journal strongly suggests displaying the actual P value in the figures.

P values in the figures are in the legend, which is acceptable, but the use of symbols (*) in the figures are not consistently used.

Figure 6A should use (*) to represent $p < 0.05$. (P= 0.01)

Figure 9E should use (*) to represent $p < 0.05$. (P= 0.013)

All other symbols in figures using (**) should represent $P < 0.01$

(Note that symbols in Figures 3-5 are used correctly)

The order of describing stats in the legend of Figure 9E should be reversed with NEC first as in the figure.

2. Subject to compliance with relevant data protection regulations, authors must make datasets and protocols available to editors and reviewers for review and to other investigators upon publication without unreasonable restrictions and in a timely manner.

The Data availability statement needs to be corrected to state: "Raw data will be made available upon reasonable request."

3. Please include the abstract figure legend with the figure legends.

Referee #1:

In my opinion, the authors have adequately responded to both reviewers' comments and suggestions and the paper should be accepted in the Journal of Physiology.

Referee #2:

The authors have satisfactorily revised the manuscript with a minor correction for revision.

END OF COMMENTS

Response to comments from Editors

Reviewing Editor:

Comments to ensure the paper complies with the Statistics Policy:

Please can the authors make sure to state all the p values in the figures rather than using symbols and n.s. This is in line with the journal policy on stats reporting.

Comments to the Author:

Thank you for taking the time to make the suggested amendments to the manuscript. There is still a small change that is required as outlined by one of the reviewers. In addition, please can the authors make sure to state all the p values in the figures rather than using symbols and n.s. This is in line with the journal policy on stats reporting.

>>Thank you for catching this. We have changed from symbols to P values in all figures.

Senior Editor:

Comments to the authors:

Thank you for submission of your revised research article to the Journal of Physiology for consideration. Referee 2 has a remaining concern that must be addressed. In addition, reporting of statistical P values in the figures, and the data availability statement require attention (see below). Please address the remaining comments from the external referee and as well as the concerns below:

1. P values:

For ease of reading and comprehension, the Journal strongly suggests displaying the actual P value in the figures.

P values in the figures are in the legend, which is acceptable, but the use of symbols (*) in the figures are not consistently used.

Figure 6A should use (*) to represent $p < 0.05$. (P= 0.01)

Figure 9E should use (*) to represent $p < 0.05$. (P= 0.013)

All other symbols in figures using (**) should represent $P < 0.01$

(Note that symbols in Figures 3-5 are used correctly)

>>To also correspond with the Reviewing Editor's request, we have removed symbols from all figures and have replaced these with P values. Thank you for spotting this.

The order of describing stats in the legend of Figure 9E should be reversed with NEC first as in the figure.

>>Thank you. We have instead changed the order of describing stats in the legend to match the figure.

2. Subject to compliance with relevant data protection regulations, authors must make datasets and protocols available to editors and reviewers for review and to other investigators upon publication without unreasonable restrictions and in a timely manner.

The Data availability statement needs to be corrected to state: "Raw data will be made available upon reasonable request."

>>We have made this change.

3. Please include the abstract figure legend with the figure legends.

>>We have included the abstract figure legend with the other legends.

Referee #1:

In my opinion, the authors have adequately responded to both reviewers' comments and suggestions and the paper should be accepted in the Journal of Physiology.

Referee #2:

The authors have satisfactorily revised the manuscript with a minor correction for revision.

>>Please note that we did not receive any specific comments for "minor correction" from this reviewer on this round. Presumably they are referring to our previous minor corrections.

Dear Dr Jonz,

Re: JP-RP-2025-287824R2 "Oxygen chemoreceptor inhibition by dopamine D2 receptors in isolated zebrafish gills" by Maddison Reed and Michael G Jonz

Thank you for submitting your manuscript to The Journal of Physiology. It has been assessed by a Reviewing Editor and by 1 expert referee and we are pleased to tell you that it is acceptable for publication following satisfactory revision.

REVISION CHECKLIST:

We look forward to receiving your revised submission.

Yours sincerely,

Harold Schultz
Senior Editor
The Journal of Physiology

EDITOR COMMENTS

Reviewing Editor:

Thank you very much for making the changes as suggested. I do sincerely apologize for you not being able to see the small but important amendment suggested by the reviewer in the previous round. Hopefully this can be viewed in the current revision and the minor amendment can be made.

Senior Editor:

Thank you for submission of your revised research article to the Journal of Physiology for consideration. For some reason the minor amendment suggested by referee 2 could not be seen by the authors in the last version. We are not sure why this occurred. Therefore this still needs to be addressed. Please address the comment here from referee 2. Thank you.

REFEREE COMMENTS

Referee #2:

Here is my 'minor comment' for the authors that was not communicated in the revised manuscript.

Acceptable after amending sentence and citation on lines 364-366 of Discussion. which currently reads.

"In many preparations D2R activation has been associated with activation of K⁺ channel currents, causing hyperpolarization, and the reduction of voltage gated Ca²⁺ currents (Vargas and Lucero, 1999)".

In the latter reference, D2R activation is NOT associated with activation of K⁺ channels, but rather with modulation/inhibition of a non-selective cation current (I_h) carried by HCN channels, permeable to both Na⁺ and K⁺. This modulation occurs via a hyperpolarizing shift in voltage dependence of I_h activation, similar to that in carotid body chemoafferent (petrosal) neurons (see Zhang et al. 2018). A better citation for this point is the paper by Einhorn et al 1991; J Neurosci 1991 Dec;11(12):3727-37, which reports DA activation of TEA-insensitive, quinine-sensitive K⁺ channels causing hyperpolarization in an endocrine cell via D2R. This appears more relevant to the present study since hypoxic depolarization of NECs is mediated by inhibition of TEA-insensitive, quinidine sensitive background K⁺ channels (Jonz et al. 2004). Also, the phrase "in many preparations" maybe overstating the case here.

END OF COMMENTS

Response to final comment from reviewer 2

Referee #2:

Here is my 'minor comment' for the authors that was not communicated in the revised manuscript.

Acceptable after amending sentence and citation on lines 364-366 of Discussion. which currently reads.

"In many preparations D2R activation has been associated with activation of K⁺ channel currents, causing hyperpolarization, and the reduction of voltage gated Ca²⁺ currents (Vargas and Lucero, 1999)".

In the latter reference, D2R activation is NOT associated with activation of K⁺ channels, but rather with modulation/inhibition of a non-selective cation current (I_h) carried by HCN channels, permeable to both Na⁺ and K⁺. This modulation occurs via a hyperpolarizing shift in voltage dependence of I_h activation, similar to that in carotid body chemoafferent (petrosal) neurons (see Zhang et al. 2018). A better citation for this point is the paper by Einhorn et al 1991; J Neurosci 1991 Dec;11(12):3727-37, which reports DA activation of TEA-insensitive, quinine-sensitive K⁺ channels causing hyperpolarization in an endocrine cell via D2R. This appears more relevant to the present study since hypoxic depolarization of NECs is mediated by inhibition of TEA-insensitive, quinidine sensitive background K⁺ channels (Jonz et al. 2004). Also, the phrase "in many preparations" maybe overstating the case here.

>>Thank you. We have made this change in the Discussion and now write: "In endocrine cells from rat, for example, D2R activation has been associated with activation of K⁺ channel currents, causing hyperpolarization, and the reduction of voltage gated Ca²⁺ currents (Einhorn et al. 1991)." We have removed the Vargas reference and have replaced it with the Einhorn paper.

Dear Dr Jonz,

Re: JP-RP-2025-287824R3 "Oxygen chemoreceptor inhibition by dopamine D2 receptors in isolated zebrafish gills" by Maddison Reed and Michael G Jonz

We are pleased to tell you that your paper has been accepted for publication in The Journal of Physiology.

Yours sincerely,

Harold Schultz
Senior Editor
The Journal of Physiology

If you would like to receive our 'Research Roundup', a monthly newsletter highlighting the cutting-edge research published in The Physiological Society's family of journals (The Journal of Physiology, Experimental Physiology, Physiological Reports, The Journal of Nutritional Physiology and The Journal of Precision Medicine: Health and Disease), please click this link, fill in your name and email address and select 'Research Roundup':
<https://www.physoc.org/journals-and-media/membernews>

- **TRANSPARENT PEER REVIEW POLICY:** To improve the transparency of its peer review process, The Journal of Physiology publishes online as supporting information the peer review history of all articles accepted for publication. Readers will have access to decision letters, including Editors' comments and referee reports, for each version of the manuscript as well as any author responses to peer review comments. Referees can decide whether or not they wish to be named on the peer review history document.
- You can help your research get the attention it deserves! Check out Wiley's free Promotion Guide for best-practice recommendations for promoting your work at: www.wileyauthors.com/eoo/guide. You can learn more about Wiley Editing Services which offers professional video, design, and writing services to create shareable video abstracts, infographics, conference posters, lay summaries, and research news stories for your research at: www.wileyauthors.com/eoo/promotion.
- **IMPORTANT NOTICE ABOUT OPEN ACCESS:** To assist authors whose funding agencies mandate public access to published research findings sooner than 12 months after publication, The Journal of Physiology allows authors to pay an Open Access (OA) fee to have their papers made freely available immediately on publication.

EDITOR COMMENTS

Reviewing Editor:

Thank you for making this final amendment. I'd again like to point out how beautiful and well presented the figures are, they are of the highest quality and clarity, and really emphasize the important messages of the study. I'm sure this work will be very well received.

Thanks again and best wishes.

Senior Editor:

The editors wish to thank the authors for these final adjustments to the manuscript. The article is now accepted for publication. Congratulations for an interesting and insightful study. Please consider the Journal of Physiology for your future studies.

REFEREE COMMENTS

Referee #2:

The authors' revisions are satisfactory and the paper is considered acceptable by this reviewer.